# GolpHCat (TMEM87A), a unique voltage-dependent cation channel in Golgi apparatus, contributes to Golgi-pH maintenance and hippocampus-dependent memory

Hyunji Kang[1,2,7], Ah-reum Han [3,7], Aihua Zhang[4,7], Heejin Jeong[5], Wuhyun Koh [1], Jung Moo Lee [1], Hayeon Lee [1], Hee Young Jo[5], Miguel A. Maria-Solano [4], Mridula Bhalla [1], Jea Kwon[1], Woo Suk Roh [1], Jimin Yang[3], Hyun Joo An[5], Sun Choi [4] ✉, Ho Min Kim [3,6] ✉ & C. Justin Lee [1,2] ✉

Impaired ion channels regulating Golgi pH lead to structural alterations in the Golgi apparatus, such as fragmentation, which is found, along with cognitive impairment, in Alzheimer's disease. However, the causal relationship between altered Golgi structure and cognitive impairment remains elusive due to the lack of understanding of ion channels in the Golgi apparatus of brain cells. Here, we identify that a transmembrane protein TMEM87A, renamed Golgi-pH-regulating cation channel (GolpHCat), expressed in astrocytes and neurons that contributes to hippocampus-dependent memory. We find that GolpHCat displays unique voltage-dependent currents, which is potently inhibited by gluconate. Additionally, we gain structural insights into the ion conduction through GolpHCat at the molecular level by determining three high-resolution cryogenic-electron microscopy structures of human GolpHCat. GolpHCat-knockout mice show fragmented Golgi morphology and altered protein glycosylation and functions in the hippocampus, leading to impaired spatial memory. These findings suggest a molecular target for Golgi-related diseases and cognitive impairment.

Golgi pH is critical for its proper morphology and function, such as the modification, packaging, and transport of proteins. Impaired Golgi pH alters protein trafficking and glycosylation[1,2] and induces morphological changes in the Golgi apparatus, such as fragmentation[3]. This morphological alteration is frequently found in neurodegenerative diseases, such as Alzheimer's and Parkinson's diseases[4–6]. Cognitive impairment is also a common symptom of these diseases[7,8]. However, the relationship between

altered Golgi structures and cognitive impairment is poorly understood.

Golgi pH has been proposed to be regulated by an adenosine triphosphate (ATP)-mediated proton pump, proton leak exchanger, proton leak channel, anion channel, and cation channel[9]. ATP-mediated proton pump and protein leak exchanger have been identified as vacuolar-type ATP hydrolase[10] and $Na^+/H^+$ exchanger (NHE)7/8[11,12], respectively. The anion channel that regulates Golgi pH

and morphology has been identified as Golgi pH regulator (GPHR)[3], which maintains normal neuronal morphology and circuitry[13]. However, the molecular identity and function of Golgi-resident cation channels in the brain remain elusive.

A potential candidate for the Golgi-resident cation channel is a transmembrane (TMEM) protein with unknown functions expressed in the plasma membrane or intracellular organelle membranes. Over the past decade, several TMEMs and their cryogenic-electron microscopy (cryo-EM) structures have been extensively investigated and found to be functional ion channels[14–20]. A TMEM with the generic name TMEM87A, identified as a predominantly Golgi-localized protein, has been proposed to play a potential role as a mechanically activated (MA) channel or an accessory subunit that modulates MA channels[21]. In a subsequent study, the cryo-EM structure of its lipid-bound form was elucidated at a resolution of ~4.7 Å[22]. However, despite these structural insights, the functionality of TMEM87A as an ion channel remains unclear due to the absence of single-channel currents in TMEM87A-reconstituted liposome patch recordings[22]. Thus, the role of TMEM87A as an ion channel remains controversial.

In this study, we identify and characterize the unique cation channel TMEM87A as a Golgi pH-regulating cation channel (GolpH-Cat), determine its atomic-level structure, delineate its ion conduction pathway, and ascertain its functions in Golgi homeostasis, protein glycosylation, and biological processes, and investigate its contribution to hippocampal spatial memory by employing multidisciplinary cutting-edge technologies, including Golgi-specific pH imaging, proteoliposome-single-channel recordings, cryo-EM structural analysis, molecular dynamic modeling, proteomics/glycomics approaches, and animal behavior tests.

## Results

### TMEM87A is localized in the Golgi apparatus and contributes to Golgi pH homeostasis in astrocytes

We first analyzed the protein sequence of full-length TMEM87A and found that TMEM87A contains a GYG sequence, which is a signature selectivity filter of classical $K^+$ channels[23] (Supplementary Fig. 1a), raising the possibility that full-length TMEM87A may be a cation channel. Full-length *TMEM87A* encodes a 63 kDa protein with a predicted N-terminal Golgi-targeting motif and seven transmembrane (TM) domains (Supplementary Fig. 1b, c). In humans, *TMEM87A* encodes three isoforms: isoform 1 is full-length with a predicted Golgi-targeting motif and TMs, isoform 2 has no TM, and isoform 3 has no predicted Golgi-targeting motif (Supplementary Fig. 1d). According to the brain RNA-seq database, full-length *TMEM87A* (isoform 1) is highly expressed in both, neurons and astrocytes[24,25]. Thus, based on bioinformatics analysis, TMEM87A is a potential candidate for the Golgi-resident cation channel in the brain.

To examine the protein expression of TMEM87A in the Golgi apparatus, we performed immunocytochemistry (ICC) against TMEM87A, using golgin-97 as a Golgi marker in cultured human astrocytes (Fig. 1a). We found that TMEM87A strongly colocalized with Golgin-97 (Pearson's correlation coefficient R: 0.75) in cultured human astrocytes (Fig. 1a), indicating that TMEM87A is mainly localized in the Golgi. To investigate whether the predicted signal sequence is indeed responsible for Golgi localization, we overexpressed individual constructs carrying the full DNA sequence each of TMEM87A isoform 1, predicted Golgi-targeting motif deleted isoform 1 (isoform 1Δ), and TMEM87A isoform 3, which we predict lacks the Golgi-targeting motif (Supplementary Fig. 1d) in cultured human astrocytes. We observed distinct and strong fluorescence signals indicating Golgi localization for isoform 1. In contrast, isoform 1Δ and isoform 3 exhibited weak fluorescence signals with different localization, even when the fluorescence intensity was saturated (Supplementary Fig. 1e). In the presence of 5 μM MG132, the expression levels of TMEM87A-iso1Δ/iso3-EGFP were increased compared to the absence of MG132

(Supplementary Fig. 1f), indicating that the Golgi-targeting motif contributes to not only Golgi localization but also protein stability by potentially increasing degradation. Further, Next-generation sequencing (RNA-seq) was used to examine the expression levels of each isoform in cultured human astrocytes (Supplementary Fig. 1g). We found that TMEM87A isoform 1 had the highest proportion of total TMEM87A expression (Supplementary Fig. 1g). Taken together, these results indicate that the major form of TMEM87A, isoform 1, is localized to the Golgi apparatus in cultured human astrocytes owing to its N-terminal signal sequence.

To examine whether TMEM87A contributes to Golgi pH, we expressed a Golgi luminal-targeting pH sensor construct, B4GALT1-RpHluorin2[26], for real-time imaging of pH in cultured human astrocytes (Fig. 1b and Supplementary Fig. 1h). We found that gene silencing of TMEM87A by shRNA led to a more basic resting Golgi pH than non-silenced (scrambled) conditions (Fig. 1c and Supplementary Fig. 1i (top)). Furthermore, Golgi pH buffering capacity, as measured by the change in pH upon 50 mM $NH_4Cl$ application, was little but statistically significantly lower in TMEM87A shRNA-transfected cells (Fig. 1d and Supplementary Fig. 1i (bottom)), indicating that TMEM87A contributes to Golgi pH buffering capacity. Taken together, these results indicate that TMEM87A, a candidate cation channel, localizes to the Golgi and contributes to Golgi pH homeostasis.

### TMEM87A mediates voltage- and pH-dependent, inwardly rectifying cationic currents

Next, to investigate whether TMEM87A mediates current in the heterologous expression system, we transfected human TMEM87A (hTMEM87A) into CHO-K1 cells, that have minimal endogenous ion channel expression, and recorded whole-cell currents under voltage-clamp conditions (Fig. 1e). Although native TMEM87A is mainly localized in the Golgi of human astrocytes, we observed that EGFP-tagged TMEM87A under heterologous overexpression was found not only in the Golgi but also in the plasma membrane (Fig. 1e), as previously reported[21]. First, we measured the voltage-dependent membrane current under a voltage-ramp protocol ranging from −150 to +100 mV with a 140 mM NaCl-containing external solution and 130 mM K-gluconate-containing internal solution (Fig. 1f). TMEM87A wild-type (WT)-mediated current displayed a non-linear current-voltage (I-V) relationship with a reversal potential near −7.7 mV and pronounced inward-rectification near −150 mV (Fig. 1f–h). The average rectification index value of TMEM87A WT from +100 to −150 mV was $2.7 \pm 0.3$ (Fig. 1i). In contrast to the WT TMEM87A-carrying cells, both outward and inward currents were completely abolished in cells carrying a mutant form of TMEM87A with the pore GYG sequence mutated to Ala-Ala-Ala (TMEME87A-AAA) (Fig. 1f–i). We performed a surface biotinylation assay and confirmed the surface expression of both TMEM87A WT and AAA mutant forms (Supplementary Fig. 2a). These results indicate that the GYG sequence in TMEM87A may play a critical role in mediating the current. Furthermore, we recorded currents with voltage-step pulses from +100 to −150 mV and found that TMEM87A-mediated currents displayed voltage-dependent inward rectification with no time- or voltage-dependent inactivation (Fig. 1j). Collectively, TMEM87A mediates voltage-dependent inwardly rectifying membrane currents in a heterologous expression system.

To investigate whether TMEM87A-mediated inward currents were carried by $Na^+$ ions, we replaced $Na^+$ with *N*-methyl-D-glucamine (NMDG) (Fig. 1k). The inward current was mostly abolished, suggesting that TMEM87A may be a $Na^+$-permeable cation channel (Fig. 1k–m). To determine the relative permeability ratio of TMEM87A-mediated currents to different cations such as $K^+$ and $Cs^+$, we replaced $Na^+$ with $K^+$ or $Cs^+$ (Fig. 1k–m) and found that reversal potentials were slightly shifted to more positive potentials: from $Na^+$

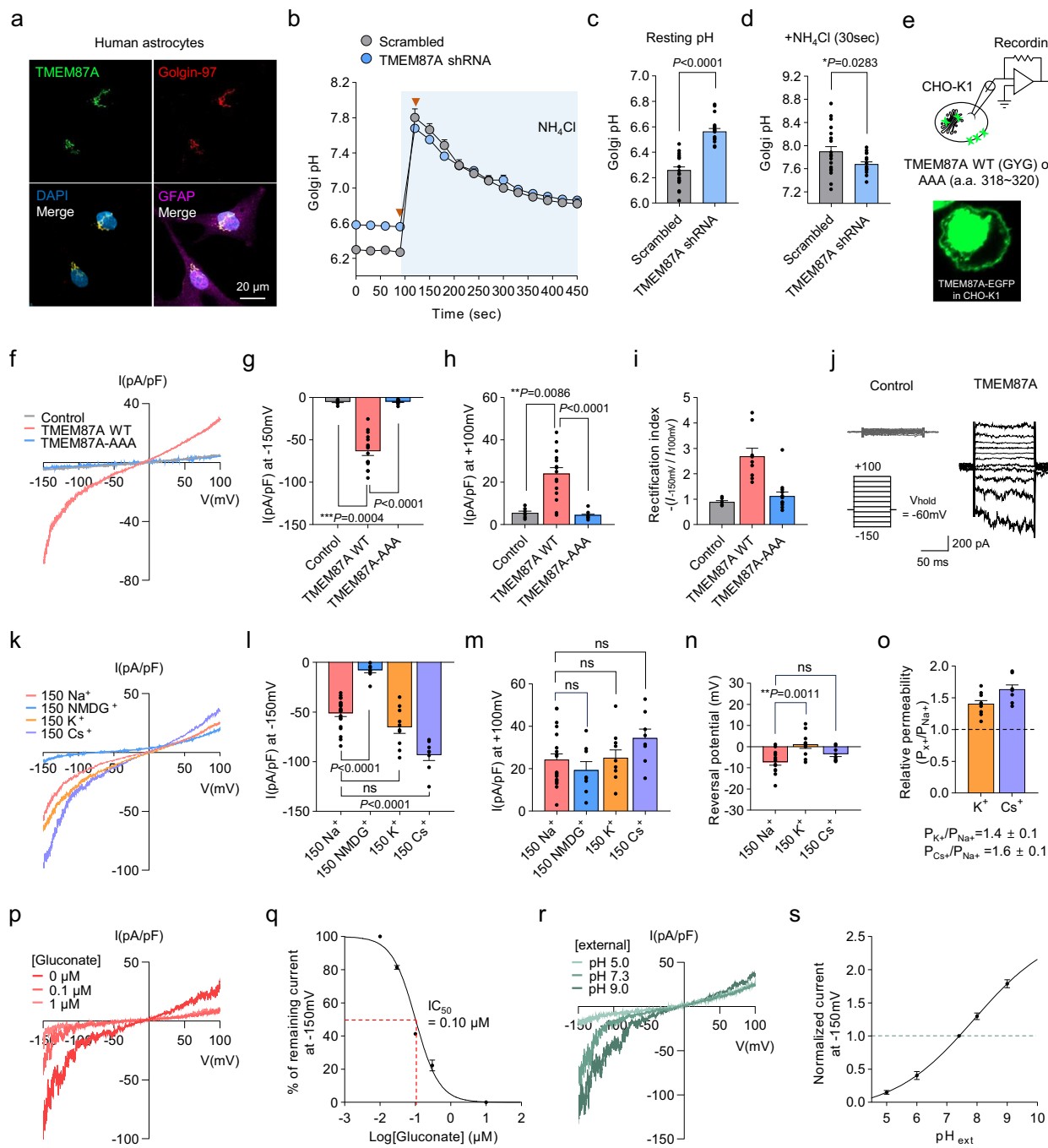

$(-7.7 \pm 1.5 \, mV)$ to $K^+$ $(0.5 \pm 1.6 \, mV)$ and $Cs^+$ $(-3.5 \pm 1.1 \, mV)$ (Fig. 1n). Using a modified Goldman–Hodgkin–Katz equation, we calculated the permeability ratios to be $P_{K+}/P_{Na+} = 1.4 \pm 0.1$ and $P_{Cs+}/P_{Na+} = 1.6 \pm 0.1$ (Fig. 1o), suggesting that TMEM87A might be a nonselective cation channel with a slightly higher permeability to $K^+$ and $Cs^+$ compared to $Na^+$. We further confirmed that TMEM87A-mediated current was not carried by $Cl^-$ (Supplementary Fig. 2b, c). Taken together, these results provide a series of evidence that TMEM87A might be a nonselective cation channel.

To investigate the pharmacological properties of TMEM87A-mediated currents, we tested the inhibitory effects of gadolinium ($Gd^{3+}$), a well-known nonselective cation channel blocker. We found that $Gd^{3+}$ effectively blocked TMEM87A-mediated currents in a dose-dependent manner with a half-maximal inhibition ($IC_{50}$) of $2.98 \, \mu M$ (Supplementary Fig. 2d, e). Similar to hTMEM87A, mouse TMEM87A

mediated similar magnitudes of membrane currents with similar gadolinium sensitivity (Supplementary Fig. 2f, g). While we were substituting $Cl^-$ with various large anions, we serendipitously discovered that gluconate potently blocked TMEM87A-mediated currents with $IC_{50}$ of $0.10 \, \mu M$ (Fig. 1p, q).

To investigate if the TMEM87A-mediated current was sensitive to pH, we recorded currents in bath solutions having different pH values (Fig. 1r). We found that the TMEM87A-mediated inward currents were significantly reduced at acidic pH, whereas they were significantly enhanced at basic pH (Fig. 1s), indicating that TMEM87A-mediated current was pH-sensitive. Despite its pH sensitivity, TMEM87A exhibited negligible permeability to $H^+$ (Supplementary Fig. 2h, i). Taken together, these results suggest that TMEM87A might be a voltage- and pH-dependent, nonselective, and inwardly rectifying cation channel.

**Fig. 1 | TMEM87A regulates Golgi pH in human astrocytes and mediates voltage- and pH-dependent, inwardly rectifying cationic currents in CHO-K1 cells. a** Colocalization of TMEM87A with the Golgin-97, Golgi marker, in cultured human astrocytes. **b** Comparison of resting Golgi luminal pH and buffer capacities under the absence and presence of 50 mM $NH_4Cl$ in Scrambled (gray, $n = 14$ cells) or TMEM87A shRNA (blue, $n = 18$ cells) transfected cultured human astrocytes expressing B4GALT1-RpHluorin. Arrows indicate time points in (**c**, **d**). **c** Golgi resting pH values. **d** Golgi pH values after treating the 50 mM $NH_4Cl$ at 30 s. **c**, **d** $n = 21$ cells for Scrambled and $n = 18$ cells for TMEM87A shRNA. **e** Schematic diagram of whole-cell patch-clamp recording from TMEM87A WT or TMEM87A-AAA (a.a. 318–320) transfected CHO-K1 cell. Inset: fluorescence image of EGFP-tagged TMEM87A. **f** Averaged I-V relationship from Control (gray), TMEM87A WT (pink), or TMEM87A-AAA (blue) transfected cells under voltage-ramp protocol (from +100 to −150 mV). **g, h** Current densities measured at −150 mV in (**g**) and +100 mV in (**h**). **i** The rectification index is calculated as the absolute ratio of amplitude at −150 mV over at +100 mV. **g–i** $n = 27$ cells for Control, n = 16 cells for TMEM87A WT, or $n = 12$ cell for TMEM87A-AAA. **j** Representative currents from Control and TMEM87A WT transfected cells under voltage-step protocol (from +100 mV to −150 mV, 25 mV step). **k** Averaged I-V relationship from TMEM87A WT transfected cells with bath solutions containing $Na^+$, $NMDG^+$, $K^+$, or $Cs^+$. **l, m** Current densities measured at −150 mV in (**l**) and +100 mV in (**m**). **k–m** $n = 21$ cells for $Na^+$, $n = 8$ cells for $NMDG^+$, $n = 10$ cells for $K^+$, $n = 8$ for $Cs^+$. **n** Reversal potentials measured from TMEM87A WT transfected cells with bath solutions containing $Na^+$ ($n = 13$ cells), $K^+$ ($n = 10$ cells), or $Cs^+$ (n = 8 cells). **o** Relative permeability ratio of $Na^+$ to $K^+$ ($P_{K+}/P_{Na+}$, $n = 10$ cells) or $Cs^+$ ($P_{Cs+}/P_{Na+}$, $n = 8$ cells). **p** Representative I-V relationship from TMEM87A WT transfected cell with or without gluconate in the bath solution. **q** Dose-response curve for percentage currents at −150 mV for gluconate concentrations (n = 5 cells). **r** Representative I-V relationship from TMEM87A WT transfected cell under various pH. **s** Normalized currents at −150 mV under various pH, normalized to current at pH 7.3 (n = 7 cells). Data were presented as the mean ± SEM. Statistical analyses were performed using two-tailed unpaired *t*-test in (**c**) ($t = 8.453$, df = 37); two-tailed unpaired *t*-test with Welch's test (**d**) ($t = 2.189$, df = 37); Kruskal–Wallis test followed by Dunn's multiple comparisons test in **g** ($H = 25.34$), **h** (H = 21.98), **i** ($H = 15.61$); one-way ANOVA followed by Dunnett's multiple comparisons test in **l** ($F(3,43) = 44.10$), **m** ($F(3,43) = 2.164$), and **n** ($F(2,28) = 7.552$). Source data and exact *p* values are provided as a Source Data file.

## TMEM87A is a bona fide ion channel

To investigate whether TMEM87A is a bona fide functional ion channel, we performed blister-attached patch recordings to study single-channel activity with 200 mM KCl symmetric solutions using reconstituted TMEM87A proteins in liposomes (proteoliposomes) (Fig. 2a). For liposome preparation, we employed a combination of both neutral and negatively charged lipids, such as 1-palmitoyl-2-oleoyl-*sn*-glycero−3-phosphocholine (POPC) and palmitoyloleoylphosphatidylglycerol (POPG), which distinguishes our approach from a previous study[22] that reported a lack of single-channel activity using only soy phosphatidylcholine (PC) (Fig. 2a). Full-length hTMEM87A (M1 ~ E555) with a cleavable C-terminus EGFP-tag and a Twin-strep tag was expressed in Expi293F cells, solubilized, purified with *n*-dodecyl β-D-maltoside (DDM) and cholesteryl hemisuccinate (CHS), and reconstituted into proteoliposome for the analysis of in vitro channel activity (Supplementary Fig. 2j and Methods).

TMEM87A in proteoliposome displayed conspicuous stochastic single-channel openings at positive or negative holding potentials from +90 to −150 mV, but not at 0 mV (Fig. 2b). A detailed analysis revealed that the channel's open probability (Po) starts from Po = 0 at 0 mV and increases non-linearly at both negative and positive potentials, with a maximum Po ≈ 0.6 at +90 mV and Po ≈0.3 at −150 mV (Fig. 2c). These results indicate that purified TMEM87A is a voltage-dependent channel with activation voltages at both negative and positive potentials and a much higher Po at positive potentials (sixfold higher Po at +90 mV compared to Po at −90 mV; Fig. 2c). The analysis of amplitude showed weak inward rectification (Fig. 2d), which was in marked contrast to the strong inward rectification observed in the whole-cell patch results (Fig. 1f). However, when we multiplied the unitary current and open probability at each holding potential, we obtained a strongly rectifying I-V relationship (Fig. 2e), implying that the whole-cell patch results originated from the ensemble average of the single-channel activities of TMEM87A. Interestingly, TMEM87A showed no subconductance opening at negative potentials, including −150 mV, whereas there were numerous subconductance-level openings ($O_{sub}$) at +90 mV (Fig. 2b, f). In addition, we found that the single-channel conductance gradually increased at negative potentials and gradually decreased at positive potentials (Fig. 2b). These results indicate that the voltage-dependent gating and ion permeation of TMEM87A are profoundly different at positive and negative potentials. The open- and closed-time distribution plots showed that TMEM87A frequently opened and closed for brief periods at +90 mV with time constants $\tau_{open} = 26$ ms and $\tau_{close} = 41$ ms (Supplementary Fig. 2l, m), whereas it less frequently opened and closed for longer periods at −150 mV with time constants $\tau_{open} = 421$ ms and $\tau_{close} = 505$ ms (Supplementary

Fig. 2n, o), again indicating a profoundly different channel gating at positive and negative potentials. Using liquid chromatography-mass spectrometry (LC-MS), in the purified TMEM87A solutions, we found negligible amounts of other pore-forming ion channel proteins (Supplementary Fig. 2k and Supplementary Table 1), which could have confounded our conclusion that TMEM87A is a bona fide ion channel. Taken together, these results provide direct evidence that TMEM87A is a voltage-dependent cation channel.

## Overall structure of hTMEM87A

To address the molecular mechanism of hTMEM87A as a voltage-dependent cation channel, we determined the structure of hTMEM87A at an overall resolution of 3.1 Å (Fig. 3a, Supplementary Fig. 3, and Supplementary Table 2) using single-particle cryo-EM in the detergent condition. In the final density maps, we reliably assigned most of the side chains (D38-P473) as well as three N-linked oligosaccharides (N62, N79, and N127), except for two loops (L148–K167 and S193–L202) and the C-terminal tail (L474-E555), probably due to their structural flexibility (Fig. 3b, c and Supplementary Fig. 3j–l).

Contrary to our expectation that TMEM87A would form a tetramer similar to other $K^+$ channels, such as KcsA and HCN channels containing a GYG motif[23,27], our data revealed that hTMEM87A is a monomer, not a tetramer (Fig. 3a). The structure of hTMEM87A contains two distinct domains: the globular domain [termed the extracellular/luminal domain (ELD), D38-K213] containing three glycans, N62, N79, and N127, and the transmembrane domain (TMD, Y224-P473) having seven transmembrane helices (7TM) (Fig. 3b, c and Supplementary Fig. 4a, b). The 7TM of the TMD are arranged counterclockwise and connected by three intracellular loops (ICL1-ICL3) and three extracellular/luminal loops (ELL1-ELL3) (Fig. 3b, c). In the TMD, TM1, TM6, and TM7 form a flat plane perpendicular to the lipid membrane (hereafter referred to as the TM plane). ELD has broad interactions with the top edges of the flat TM plane and the extracellular loop connecting TM2 and TM3 (ELL1) (Supplementary Fig. 4d, e; interaction patch 1–4). Combined with the inter-domain disulfide bond (C89–C431), these broad interactions are likely to restrain ELD movement, thus maintaining a fixed orientation of the ELD relative to the TM plane.

Since voltage-dependent inwardly rectifying currents were detected in the proteoliposomes reconstituted with hTMEM87A (Fig. 2), we were intrigued to find a central cavity buried deep in the TMD for ion conduction. Two cavities are located between the TM plane and the tilted/twisted TM2-TM5 helices; an upper hydrophilic cavity (cyan dashed circle) and a lower hydrophobic cavity (yellow dashed circle) (Fig. 3d). The upper hydrophilic cavity opens to the luminal side near the ELD and is exposed to the upper leaflet of the

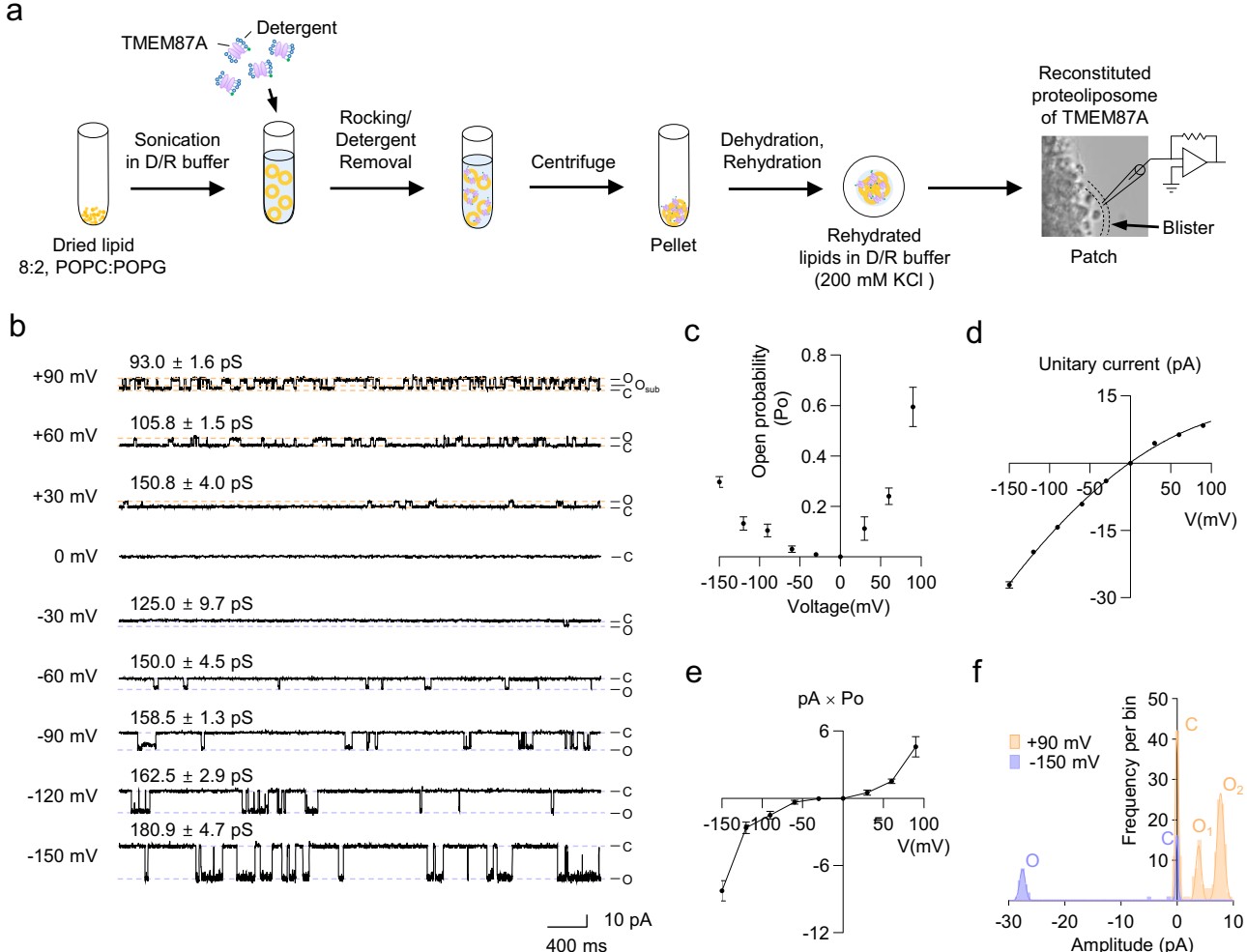

**Fig. 2 | TMEM87A is a bona fide functional ion channel in proteoliposome.**
**a** Schematic diagram showing the procedure for reconstitution of TMEM87A into liposome (8:2, POPC:POPG) with dehydration/rehydration method for single-channel recording. **b** Representative spontaneous single-channel currents from the reconstituted proteoliposome of TMEM87A under voltage steps (from +90 to −150 mV, 30 mV step) from the same patch condition. **c** Voltage-dependent channel-opening probability (Po) of TMEM87A at each holding potential ($n = 3$). **d** I-V relationship of TMEM87A single-channel unitary current activities ($n = 3$). Data were fitted with a polynomial. **e** The unitary current × open probability-voltage relationship of TMEM87A ($n = 3$). **f** Amplitude histogram of TMEM87A single-channel unitary current activities with open (O) and closed (C) states at +90 mV (orange) and −150 mV (purple) from (**b**). Distribution data are fitted with a sum of two Gaussians at each holding potential. Data were presented as the mean ± SEM. The $n$ numbers are from three independent proteoliposome experiments. Source data are provided as a Source Data file.

lipid bilayer through a gap between TM5 and TM6. Compared to the previously reported hTMEM87A structure (PDB ID: 8CTJ)[22], we observed a well-resolved electron density in the TMD pocket, while overall architecture is similar (overall root mean square deviation (RMSD) = 1.03 Å, Supplementary Fig. 3i–k). Based on the shape of the electron density and the lipid composition of the Golgi apparatus membrane[28], we modeled phosphatidylethanolamine (PE, 1-palmitoyl-2-oleoyl-*sn*-glycerol-3-phosphoethanolamine, or PE-16:0-18:1) to this density (Fig. 3b and Supplementary Fig. 3k). Although no phospholipids had been added during protein purification, and the protein was dissolved in LMNG/CHS detergent, PE was co-purified with hTMEM87A, suggesting that PE tightly and stably binds to TMEM87A. Indeed, the lateral side of the central cavity of the TMD was partially sealed by PE, of which the R2-fatty acid chain fully occupied the lower hydrophobic cavity (Fig. 3d). In particular, the terminal amine of PE formed a charge-interaction with TM4 E347 and TM7 W445, and its phosphate group was coordinated by R305 and R309 of TM3, Y340 of TM4, and D371 of TM5 (Fig. 3e). In addition, the oxygen atoms in the ester group were stabilized by TM5 D371 and TM6 S415. Side chains of TM helices [α2 (I258, I262, and V265), α3 (A308, L311, V312, and V315),

α5 (I378), α6 (F404, L408, A411, and V412) and α7 (F449, I452, L453, and I456)] make hydrophobic interactions with the R2-fatty acid chain. The remaining R1-fatty acid chain of PE protrudes through the gap between TM5 and TM6, where it interacts with C375, W376, and F379 of TM5 and is likely to interact with other lipid molecules in the membrane bilayer. The residues participating in these extensive interactions with PE are highly conserved (Fig. 3e and Supplementary Fig. 5a), indicating a physiological role for PE in maintaining the structural and functional integrity of hTMEM87A. Taken together, these results indicate that hTMEM87A has a monomer architecture composed of an ELD and seven TMD, with a well-resolved PE in the TMD pocket.

## Comparison of hTMEM87A with its structural homologs
To investigate the structural basis of the TMEM87A ELD and TMD for their physiological roles, we initially searched for structural homology using the Dali server[29]. As discussed in the previous study[22], we found that ELD resembles a Golgi Dynamics (GOLD) domain present in p24 family proteins and SEC14-like protein 3, which are implicated in the secretory pathway such as cargo sorting and membrane trafficking (Supplementary Fig. 4f)[22,30,31]. Interestingly, the sequence identity of

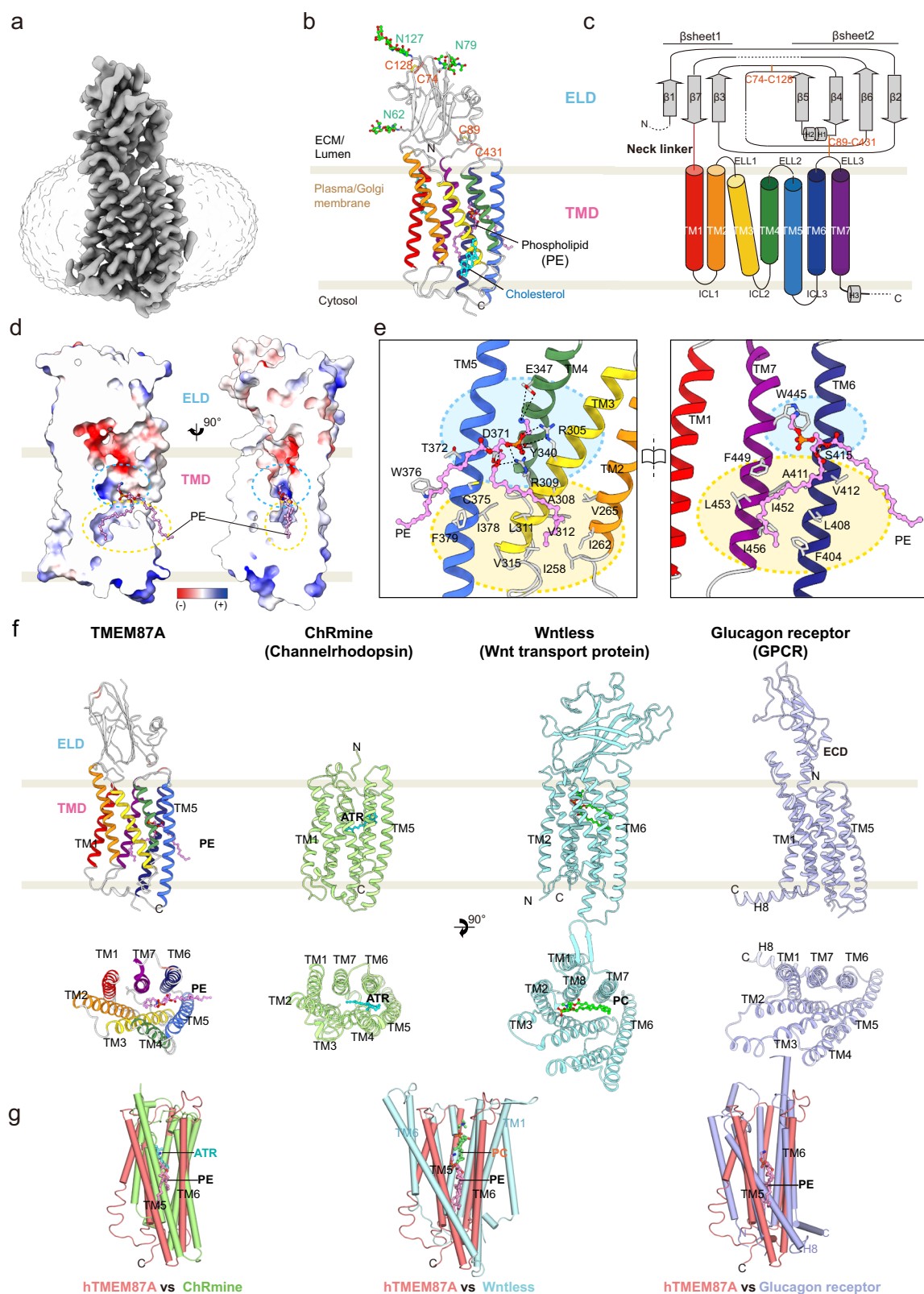

ELD across hTMEM87 family members is relatively low compared to that of the TMD (Supplementary Fig. 5b). However, the results of structure prediction with AlphaFold2[32] indicated that the ELD structure of hTMEM87A and hTMEM87B were highly similar (Supplementary Fig. 4c), suggesting functional redundancy between TMEM87 family members. Given the roles of GOLD domain-containing proteins in the secretory pathway, the ELD of the hTMEM87 family may play a

role in protein trafficking by interacting with unidentified partners. Sequence comparison of TMEM87A with eukaryote orthologs[33] suggests that the conserved evolutionary TMD is more relevant to the physiological function of TMEM87A (Supplementary Fig. 5a). From a Dali server search with hTMEM87A TMD, we found that structural homologs of hTMEM87A TMD included microbial channelrhodopsin (ChRmine[34], PDB: 7W9W, TM1-TM7 among 7TM, Z-score 13.7, RMSD

**Fig. 3 | Cryo-EM structure and structural feature of human TMEM87A. a** Cryo-EM density map of hTMEM87A, colored in slate gray. The density of the micelle (contoured at 0.161σ) is presented as light gray. **b** Overall structure of hTMEM87A, with ELD (light gray) and TMD [rainbow color from TM1 (red) to TM7 (purple)]. Disulfide bridges (orange, C74–C128, and C89–C431) and N-linked glycans (green, N62, N79, and N127) are shown as sticks. PE and cholesterol are indicated as pink and cyan sticks, respectively. **c** Topology of hTMEM87A. ELD consists of two α-helices and seven β-strands arranged in an anti-parallel β-sandwich. The secondary structure elements (cylinder for helix and arrow for strand) are colored as in (**b**). Disulfide bonds are shown as an orange line. Dashed lines denote regions where density was insufficient for model building. **d** Two different views of the vertical cross-section of the PE-binding pocket in TMD. The electrostatic surface potential of the central cavity is shown. The upper hydrophilic and lower hydrophobic cavities are indicated as cyan and yellow dashed circles, respectively. **e** Open-book views of the PE-binding pocket and the interaction details. Interaction residues with PE are shown as sticks. The hydrogen and ionic bonds are depicted as a dashed line. Cyan and yellow colored circles represent hydrophilic and hydrophobic cavities in TMD, respectively. **f** Structural comparison of hTMEM87A TMD with other seven transmembranes (7TM) proteins ChRmine (PDB:7W9W, pale green), Wntless (PDB:7DRT, cyan), and Glucagon receptor (PDB:5YQZ, light purple), shown as a side view (top) and top view (bottom). In the top view, ELD (hTMEM87A), luminal domain (LD, Wntless), and extracellular domain (ECD, glucagon receptor) are omitted for clarity. PE in hTMEM87A, ATR in ChRmine, and Phosphatidylcholine (PC) in Wntless are displayed as pink, cyan, and lime sticks, respectively. **g** Superimposition of hTMEM87A TMD and TM region of ChRmine (pale green), Wntless (cyan), or glucagon receptor (light purple). The view is a 90° rotation view of (**b**). along the y-axis to show the lateral opening between TM5 and TM6 of hTMEM87A TMD.

3.3 over 214 residues), Wnt transport protein (Wntless[35], PDB: 7DRT, TM2-TM8 among eight TM helices, Z-score 14.2, RMSD 4.1 over 233 residues), and glucagon G-protein coupled receptor (Glucagon receptor[36], PDB: 5YQZ, TM1-TM7 among eight TM helices, Z-score 11.7, RMSD 4.0 over 223 residues) (Fig. 3f), although their sequence identity is relatively low (8–13%). hTMEM87A is similar to Wntless and glucagon receptors in having a central cavity buried in the TMD located below the globular extracellular domain (ECD) and opening to the luminal side. However, the position of the ECDs differs substantially from that of the hTMEM87A ELD. Moreover, TM4 and TM5 of hTMEM87A are tilted toward the cavity core, resulting in a smaller central cavity compared to the cavities of Wntless and glucagon receptors, which can accommodate a lipidated Wnt3a/8a hairpin and a peptide ligand, respectively (Fig. 3f, g). The overall arrangement of hTMEM87A 7TM was much closer to that of ChRmine, although it does not have an ECD. ChRmine and hTMEM87A are superimposed with an overall RMSD of 3.3 Å over 7TM (214 residues) and two of their corresponding TMD layers (TM1/6/7 and TM2/3/4/5) are more tightly packed with each other than those of Wntless and glucagon receptor (Fig. 3g). Similar to all-trans-retinal (ATR) in ChRmine, PE is surrounded by TM3-TM7 in hTMEM87A, although their binding orientations are quite different (Fig. 3f, g). Taken together, the 7TM of hTMEM87A is structurally similar to ChRmine, even though its extracellular globular domain resembles the GOLD domain.

## Putative Ion conduction pathway of hTMEM87A

Next, we examined the configuration of the hTMEM87A cavity to delineate the location and shape of the ion conduction pathway. No continuous channel pores were observed in the structure (Fig. 4a). The water-accessible cavity was physically blocked by the R2-fatty acid chain of PE, which extends from the lateral opening between TM5 and TM6 to the lower hydrophobic cavity. These data suggested that the observed conformation was presumably in a closed state. However, analysis of the electrostatic surface potentials of hTMEM87A revealed that negatively charged residues (D38 in ELD and E222, D223, E279, E298, D441, and D442 in TMD), main-chain carbonyl groups of L438 and W439, and the hydroxyl group of ELD Y90 are distributed on the funnel-shaped luminal vestibule (Fig. 4a–c; hereafter negatively charged luminal vestibule (NLV)). Similar to the extracellular vestibule of ChRmine[34], the electronegative surface potential of hTMEM87A NLV may attract and stabilize positively charged ions, thereby effectively increasing the local cation concentration. To examine the potential role of these negatively charged NLV residues in ion conduction, we performed mutagenesis and measured hTMEM87A channel activity using whole-cell patch clamping. Among the three negatively charged residues, the E279A mutation resulted in almost complete elimination of channel activity, whereas the E298A mutation showed partially decreased channel activity, and the D442A mutation showed no change compared to WT (Fig. 4d and Supplementary Fig. 6a, b). Further investigation of the role of the NLV in cation attraction with

accelerated Gaussian molecular dynamics (GaMD) simulations[37,38] revealed that potassium cations (K⁺) are recognized by the superficial region of the NLV (D38, Y90, E222, and D223), stabilized in deeper NLV regions upon interaction with additional residues (D441 and D442), and are able to travel a short distance through the channel (Fig. 4e and Supplementary Fig. 6g). In fact, K⁺ binding events occur frequently, as we explored up to 12 bindings within the 1.5 µs GaMD simulations. Collectively, the attraction of the cation for ion conduction was orchestrated by the negatively charged residues on the funnel-shaped luminal vestibule, whereas complete cation permeabilization was not observed, possibly because we started with a closed structure, and the expected timescale of the channel pore opening is long (-10 ms).

It has been shown that gluconate can effectively block both outward and inward currents of hTMEM87A with 0.10 µM IC₅₀ (Fig. 1p, q). Thus, to decipher the binding pocket for gluconate and the putative ion conduction pathway of hTMEM87A, we incubated purified hTMEM87A with 10 mM sodium gluconate and determined the cryo-EM structure of hTMEM87A-Gluconate (hTMEM87A-Gluc) at -3.6 Å resolution (Supplementary Fig. 3m–t). While the overall structure of hTMEM87A-Gluc was essentially identical to hTMEM87A structure, of potential significance, the observed density indicated that a gluconate ion occupied the hydrophilic cavity of hTMEM87A via electrostatic interactions with R305, R309, D371, and W445, in a manner similar to the head group of PE (Fig. 4f, g and Supplementary Fig. 3t). Based on these observations, we hypothesized that the extended electrostatic-interaction networks (Y237, E272, K273, S301, K304, R305, R309, S344, D371, Y340, and S415) underneath the NLV, while also mediating the binding of the PE head group, can form a constriction and hence, be implicated in hTMEM87A-mediated ion conduction upon the channel-opening stimulus. To test this hypothesis, we expressed hTMEM87A mutants (Y237A, E272A, K273A, S301A, K304A, R305A, and R309A) and recorded their channel activity. Despite their robust cell membrane expression, current amplitudes at −150 mV were significantly decreased for all mutants compared to WT (Fig. 4h and Supplementary Fig. 6a, d, f). Moreover, these residues in the constriction sites are highly conserved (Supplementary Fig. 5a). These data suggest that a channel-opening stimulus (high voltage in our experiments) can trigger rearrangements in the electrostatic-interaction networks underneath the NLV, ultimately leading to ion conduction. Conceivably, such a mechanism resembles that of channelrhodopsin photocurrents, which are initiated by retinal isomerization and drive conformational changes in transmembrane helices[39–41].

## Role of phosphatidylethanolamine and TM3 in ion conduction of hTMEM87A

Next, we asked why endogenous PEs bind to TMEM87A, despite phosphatidylcholine (PC) being the most abundant lipid in the Golgi membrane (PC, -50%; PE, -20%)[28]. To this end, we compared the binding free energies of PC and PE to hTMEM87A using MD simulations and assessed them using a linear interaction energy (LIE) model[42]

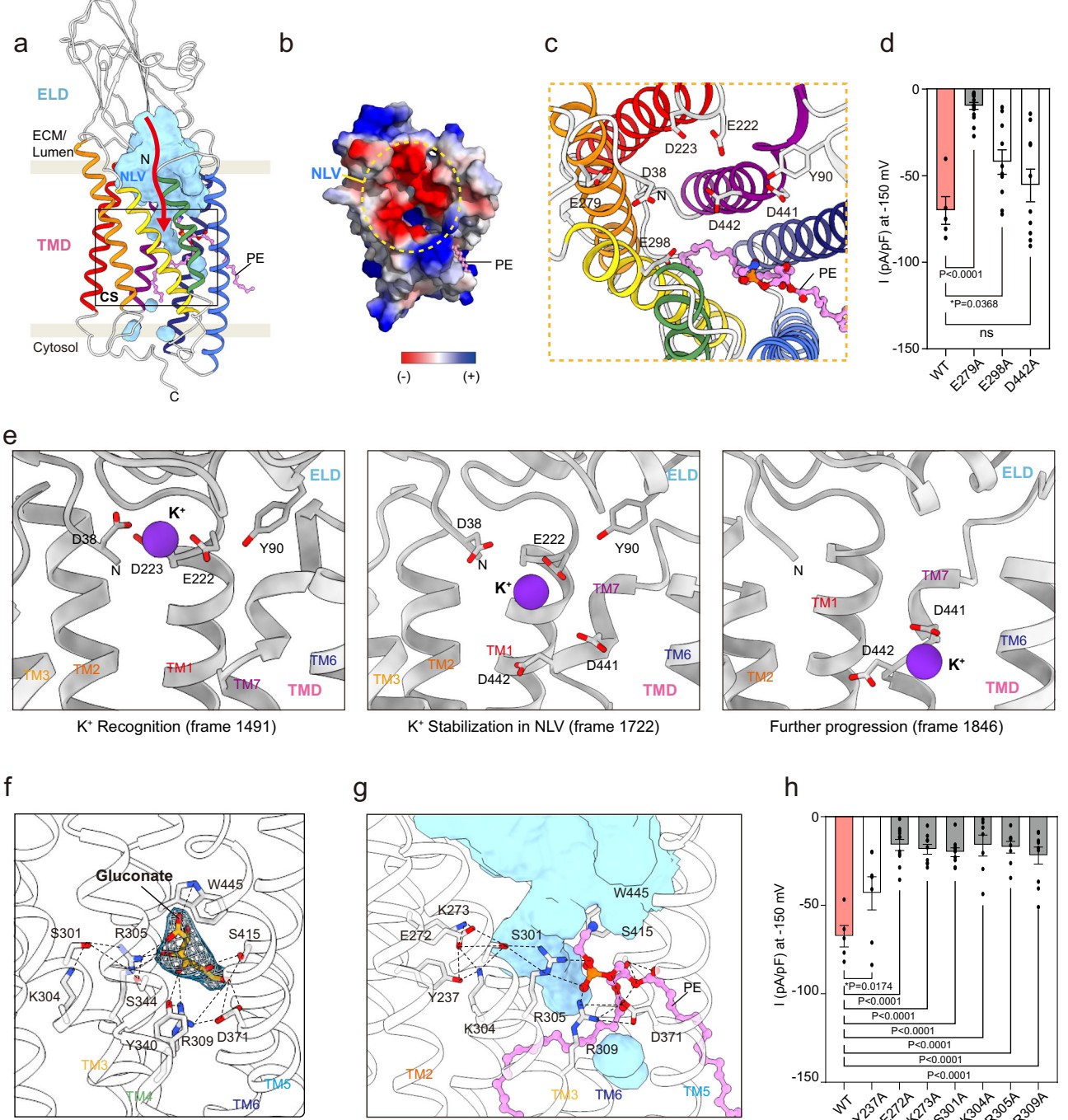

**Fig. 4 | Putative ion-conducting pathway in hTMEM87A. a** Organization of ion-pathway in hTMEM87A. Water-accessible cavities are shown as a cyan surface, with the putative ion conduction pathway indicated by a red arrow. Negative-charged luminal vestibule (NLV) and constriction site (CS, black-lined box) are labeled. **b** The surface electrostatic potential of the NLV (yellow dotted circle). ELD and ten residues (R351-S361) are omitted for clarity. **c** Close-up views of NLV and key negative-charged residues are shown as sticks. **d** Current density measured at −150 mV for hTMEM87A WT ($n = 5$) and NLV mutants ($n = 15$ for E279A, $n = 10$ for E298A, and $n = 10$ for D442A). **e** Representative structures of the K+ conformational dynamics in the NLV of hTMEM87A obtained from GaMD simulations 1. K+ atoms (purple sphere) and their interacting residues (light gray stick) are displayed. TM4 and TM5 are omitted for clarity. **f** Close-up view of gluconate binding site in hTMEM87A. Gluconate (yellow stick) with cryo-EM density map (contour level =

0.118) and interaction residues (gray sticks) are shown. The hydrogen and ionic bonds are depicted as a dashed line. Helices of hTMEM87A TMD are displayed as transparent cartoons. **g** Close-up views of the constriction site and the interaction details. Key interaction residues (gray) and PE (pink) are shown as sticks. Cavities are shown as cyan surfaces. Helices of hTMEM87A TMD are displayed as transparent cartoons. TM4 is omitted for clarity. **h** Current density measured at −150 mV for hTMEM87A WT ($n = 5$) and CS mutants ($n = 7$ for Y237A, $n = 11$ for E272A, $n = 9$ for K273A, $n = 9$ for S301A, $n = 7$ for K304A, $n = 8$ for R305A, and $n = 10$ for R309A). Data were presented as the mean ± SEM. Statistical analyses were performed using student one-way ANOVA followed by Dunnett's multiple comparisons test in **d** ($F(3,36) = 16.64$) and **h** ($F(7,58) = 10.84$). Source data and exact $p$ values are provided as a Source Data file.

(Supplementary Fig. 6h–l). The results showed that the additional methyl groups of PC, while providing greater van der Waals interaction energy, also reduce the coulombic interaction energy due to their screening effect. However, the calculated binding free energy of PC to hTMEM87A is higher than that of PE by 19.2 kJ/mol ($\Delta F_{m\rightarrow p}$ (PC) = −13.2 kJ/mol and $\Delta F_{m\rightarrow p}$ (PE) = −32.4 kJ/mol), suggesting that under physiological condition, PE is more likely to bind to hTMEM87A. Importantly, these data support that the PE observed in the cryo-EM structure was co-purified with hTMEM87A from the Golgi membrane. However, future studies employing techniques like mass spectrometry could further strengthen our conclusions and provide complementary evidence for PE in TMD. To characterize the entry of PE and binding to hTMEM87A, we simulated five trajectories up to 1 µs starting from a system (named $S_{p^*/L}$) constructed by removing the bound PE lipid (L) from the hTMEM87A structure and then embedding the bare hTMEM87A (p*) in a lipid bilayer composed of only PE (Fig. 5). To represent the binding process, we defined two distances, dP and dR2-Cent, as those from the phosphorus atom and the center of mass of the R2-fatty acid chain of PE to the smallest-moment principal axis (Pz) of TMD, respectively (Fig. 5a). In the two-dimensional histograms of dP and dR2-Cent for all five trajectories, we identified seven highly populated states of PE (from S1 to S7), among which the fully bound state (S1) was the most probable. The conformational snapshots corresponding to these seven states are shown in Fig. 5c. In one example trajectory, for which variations in dP and dR2-Cent as a function of time are presented, the PE lipid starts entering the TMD cavity quickly (~25 ns), passes through two intermediate partially bound states (S3 and S4), and arrives at the fully bound state (S1) after ~500 ns (Fig. 5b). These data demonstrate that lipid entry occurs in a stepwise manner. Moreover, the proximity between the state corresponding to the cryo-EM structure (S*) and S1 indicates a clear tendency for PE to be inserted at this unique position of hTMEM87A.

Although we investigated the ion conduction of TMEM87A, we could not identify an ion conduction pathway beyond the constriction site due to the presence of the R2-fatty acid chain of PE occupying the hydrophobic TMD cavity (Fig. 3d, e). Previous studies have suggested that hydrophobic regions in ion conduction pathways, such as those found in KcsA[43] and TWIK-1[44], contribute to ion permeation and gating processes[45,46]. Based on this, we hypothesized that the lower hydrophobic cavity of TMEM87A might be a potential pathway for ion conduction and that our structure represents a non-conducting state in which the lipid chain obstructs ion conduction. To verify this, we determined the cryo-EM structure of a hTMEM87A mutant A308M, which we predict would block the ion conduction pathway at the entrance of the lower hydrophobic cavity (Fig. 5d and Supplementary Fig. 7). Indeed, A308M effectively sealed the lower hydrophobic cavity, resulting in reduced channel activity compared to that of the WT (Fig. 5e, f and Supplementary Fig. 6c). Interestingly, the PE density in hTMEM87A A308M was displaced to a position similar to that of the S6 state in MD simulation (Fig. 5g and Supplementary Fig. 7i). This suggests that the lower hydrophobic cavity of TMEM87A is a potential ion conduction pathway, and that PE displacement in hTMEM87A by the external energy, such as high voltage and mechanical pressure steps, could affect channel opening.

We demonstrated that the channel activity of hTMEM87A was regulated in a voltage-dependent manner (Figs. 1, 2). Voltage-regulated ion channels, such as $K_v$ channels, $Na_v$ channels, and trimeric intracellular cation (TRIC) channels, usually have a voltage-sensing domain (helix) wherein positively charged lysine or arginine residues are enriched[47–50]. The TRIC-B1 channel contains three conserved basic residues in its TM4 helix. These interact with the phosphate group of phosphatidylinositol-4,5-biphosphate (PIP$_2$) and nearby negatively charged residues, which occlude its ion permeation pathway (Fig. 5h). Moreover, the artificial disulfide bonds that lock the TM4 helix of TRIC-B1 in a restrained conformation cause it to remain closed upon

depolarization, demonstrating an essential mechanism by which the voltage-sensing TRIC TM4 helix is coupled to channel activation[48]. Although the structure of hTMEM87A is different from that of TRIC channels, three conserved basic residues (K304, R305, and R309) lining the hTMEM87A TM3 interact with an electronegative PE head group and the neighboring E272 residue, which closely resembles the voltage-sensing TM4 helix of TRIC-B1[47] (PDB: 5EGI) (Fig. 5h). Indeed, electrophysiological analysis with mutations of these residues (K304A, R305A, and R309A) showed reduced channel activity (Fig. 4h). In addition, the AAA mutation of the GYG sequence of TMEM87A at the cytoplasmic end of TM3 resulted in reduced channel activity (Figs. 1e, f, 5h, i and Supplementary Fig. 6a, e). Moreover, we found that mutations in the loop residues flanking the TM3 helix (E288R on ELL1) resulted in increased channel activity than WT (Fig. 5i and Supplementary Fig. 6e). As noted, E288 interacts with nearby basic residues, maintaining the structural integrity of hTMEM87A (Supplementary Fig. 4d, e, Patch4). Replacing the negatively charged glutamate with positively charged arginine disrupts interactions with the basic residues, potentially introducing the conformational changes to TM3 and influencing channel activity. Alternatively, the mutation might alter electrostatic interactions within the TMD, leading to an environment more conductive to ion flow. To unravel the precise mechanism, further investigations are needed. Collectively, these results suggest a critical role for the hTMEM87A TM3 in influencing channel activity, probably by participating in voltage sensing and gating (Fig. 5j). However, we did not observe any conventional channel structure from the geometric pore analysis and cation permeation pathway among all simulated trajectories (~1 µs). Possible explanations for the lack of cation permeation pathway may include (1) longer simulation times (>1 µs) needed and (2) the lack of structures for open state upon the voltage stimulation. Taken together, these structural analyses suggest that TMEM87A is a monomeric ion channel with a putative ion conduction pathway that can be opened by voltage stimulation and closed by PE.

## GolpHCat is required for normal Golgi morphology in hippocampal astrocytes and neurons

Based on the electrophysiological and structural results, we renamed TMEM87A as GolpHCat to represent a Golgi-resident, pH-regulating cation channel. To characterize the in vivo functions of GolpHCat in mice, we generated a GolpHCat knockout (KO) mouse line using CRISPR/Cas9 gene editing (Supplementary Fig. 8a). *Tmem87A* of *Mus musculus* is located on the reverse strand of mouse chromosome 2 (Chr2, 120,185,793–120,234,594) and consists of 20 exons (Supplementary Fig. 8a). Two guide RNAs (gRNAs) targeting the introns flanking exon 10 were designed to delete exon 10 containing the GYG motif sequence, which made a 255 bp deletion in the gene (Supplementary Fig. 8a). To confirm the generated mouse genotypes, we performed PCR genotyping around the deletion and observed three types of genotypes: WT with one band at 484 bp, heterozygote (HT) with two bands at 484 bp and 229 bp, and homozygote (KO) with one band at 229 bp (Supplementary Fig. 8b). GolpHCat HT and KO mice were viable with no gross abnormalities, possibly due to compensation by TMEM87B.

To examine the expression pattern of GolpHCat, we performed immunohistochemistry (IHC) using antibodies against GolpHCat and Glial Fibrillary acidic protein (GFAP) or "Neuronal Nuclei" (NeuN), which are astrocytic and neuronal markers, respectively. Strong fluorescence intensity was observed in the hippocampus, but not in other brain regions (Fig. 6a and Supplementary Fig. 8c). Furthermore, we observed the absence of GolpHCat expression in GolpHCat KO mice using IHC and western blotting (Fig. 6a and Supplementary Fig. 8h). The observed GolpHCat immunoreactivity in both GFAP- and NeuN-positive cells in WT mice was virtually absent in GolpHCat KO mice (Fig. 6b, c). These results indicate that the antibody against GolpHCat is highly specific and that GolpHCat is majorly expressed in

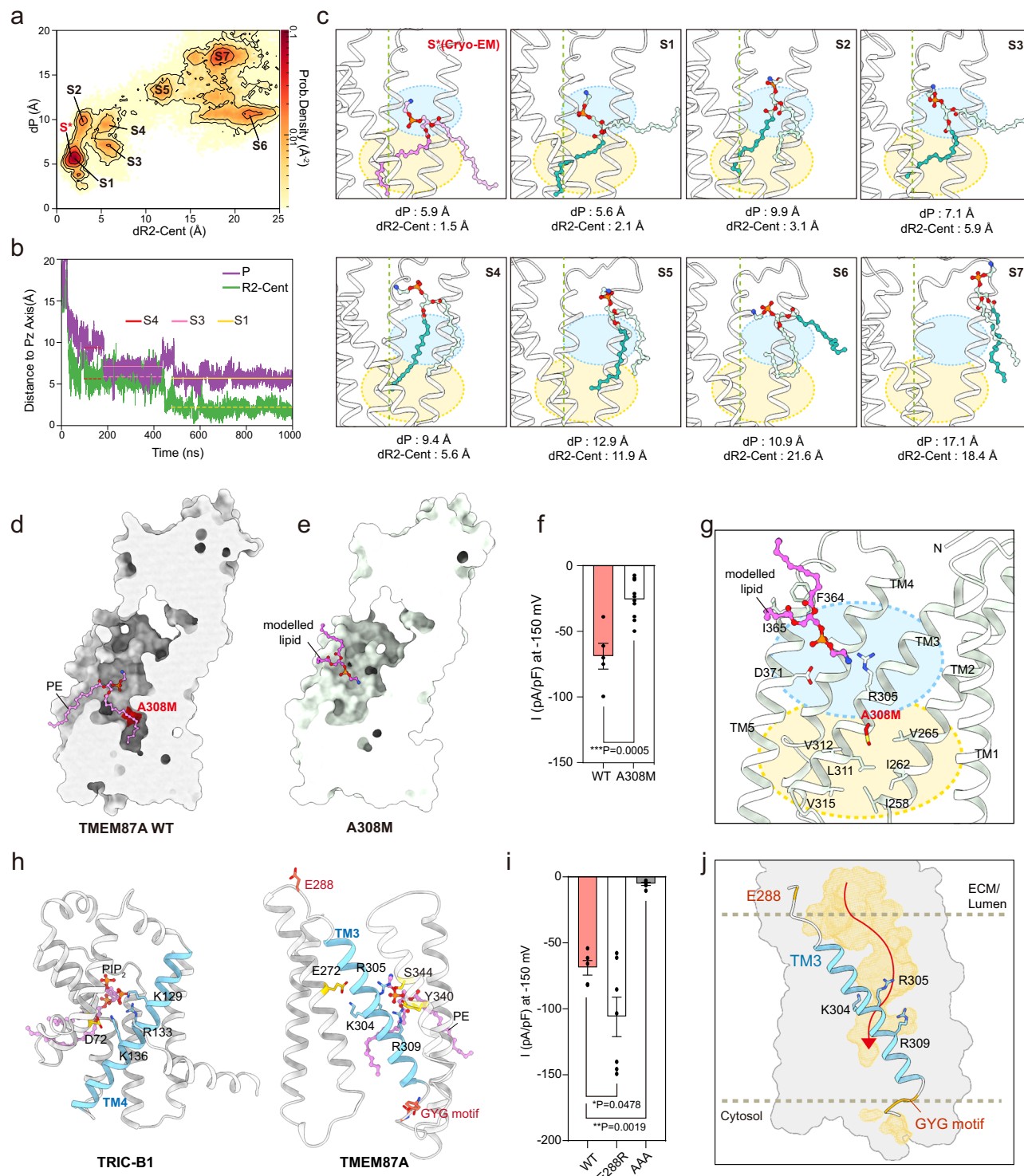

both astrocytes and neurons of the hippocampus. To investigate whether GolpHCat is localized in the Golgi of astrocytes and neurons in the mouse brain, we performed IHC with antibodies against GolpHCat and different Golgi markers (Fig. 6d), such as Golgin-97 and giantin, since both markers show different immunoreactivities in astrocytes[51], neuronal dedrites[52] and soma[53]. We found that GolpHCat was colocalized with Golgin-97 and giantin in astrocytes and soma of neurons, respectively (Fig. 6d). Pearson's correlation coefficient R of 0.5 for GFAP positive astrocytes and 0.6 for NeuN-positive neurons (Fig. 6e) indicated that GolpHCat was moderately localized in the Golgi of astrocytes and soma of neurons.

It has been previously reported that genetic ablation of the Golgi pH regulating anion channel, GPHR, disorganizes the Golgi structure with fragmentation and swelling of the cisternae[3]. To examine Golgi morphology in the hippocampal astrocytes and neurons of WT and GolpHCat KO mice, we performed transmission electron microscopy (TEM) (Fig. 6f). We observed that most of the Golgi exhibited disrupted stacks and dilated cisternae in both cell types of GolpHCat KO mice compared to WT mice (Fig. 6f). In addition, we analyzed the length of Golgi cisternae and width of Golgi apparatus (Fig. 6g) in each cell type and found that the maximum length of cisternae was significantly decreased, whereas the width of Golgi was significantly increased in

**Fig. 5 | Role of PE and TM3 in ion conduction in hTMEM87A channel activity.**
**a** 2D histogram of distances (dP and dR2-Cent) for five MD trajectories ($5 \times 1\,\mu s$ from system $S_{P^+/L}$). dP and dR2-Cent are the distances from the phosphorus atom and the center of mass of the R2-fatty acid chain to the smallest-moment principal axis (Pz) of TMD, respectively. PE in cryo-EM structure (red) and seven highly populated states of PE are labeled (from S1 to S7). **b** Variations of distances of dP and dR2-Cent as a function of time along one trajectory. States of S4, S3, and S1 are indicated by horizontal lines. **c** The conformational snapshots of PE in seven different states (S1–S7). PE from cryo-EM structure (pink) and MD simulations are displayed as sticks [the phosphorus atom (orange), R1-fatty acid chain (light blue), and R2-fatty acid chain (teal)]. Dashed lines indicate Pz of TMD. Calculated distances of P and R2-Cent are indicated. Cyan and yellow colored circles represent hydrophilic and hydrophobic cavities in TMD, respectively. **d**, **e** Cross-sectional view of hTMEM87 WT and A308M. The A308M, which blocks the PE chain from entering the inner cavity, is highlighted in red. A modeled lipid in A308M is shown as a pink stick. **f** Current density measured at −150 mV for hTMEM87A WT and A308M (bottom, $n = 5$ WT and $n = 10$ for A308M). **g** Close-up view of lipid binding site in hTMEM87A A308M. **h** Voltage-sensing TM4 of TRIC-B1 (left, PDB: 5EGI) and potential voltage sensor in TM3 of TMEM87A (right). The conserved basic residues (cyan), nearby acidic residues (yellow), and phospholipids (pink) are shown as sticks. **i** Current density measured at −150 mV for hTMEM87A WT ($n = 5$) and mutants on either ends of TM3 ($n = 7$ for E288R, and $n = 6$ for AAA). **j** Water-accessible cavities (yellow surfaces) with putative ion-pathway (red arrow). PE is omitted to show the unblocked lower hydrophobic cavity. Potential voltage-sensing helix TM3 and conserved basic residues are shown as cyan. Data were presented as the mean ± SEM. Statistical analyses were performed using a two-tailed unpaired $t$-test in (**f**) ($t = 4.605$, df = 13); one-way ANOVA followed by Dunnett's multiple comparisons test in (**i**) ($F(2,15) = 24.74$). Source data and exact $p$ values are provided as a Source Data file.

both hippocampal astrocytes and neurons of KO mice (Fig. 6h, i). This is consistent with the morphological alterations observed in GPHR KO cells[3]. In contrast, the mitochondria showed normal morphology in the cells of GolpHCat KO mice (Supplementary Fig. 8d), indicating that GolpHCat is critical for maintaining normal Golgi morphology in cells. Taken together, our results suggest that the lack of GolpHCat results in impaired Golgi homeostasis, leading to impaired Golgi functions, such as protein glycosylation.

## GolpHCat KO altered Golgi function, which affects protein glycosylation and function in the hippocampus

To investigate the glycosylation patterns, we analyzed glycans by LC-MS using hippocampal brain samples from WT and GolpHCat KO mice (Fig. 7a), as previously described[54,55]. We found that among the detected 99 N-glycans some were downregulated, while others were upregulated in the KO mice (Fig. 7a). We performed principal component analysis (PCA) and observed a complete separation of WT and KO mice by principal component 1 (Fig. 7b). Subsequently, we determined the glycan types that most strongly influenced principal component 1, highlighting the significance of C/H-F and C/H-FS glycans in our findings (Fig. 7c). Furthermore, to investigate the specific altered glycan patterns that significantly distinguished the WT and GolpHCat KO mice, we featured 14 specific glycans that displayed significant differences, as depicted in the volcano plot (Fig. 7d). Consistent with Fig. 7c, C/H-F and C/H-FS glycans exhibited significant alterations in KO samples compared to WT samples. To gain further insight into the changes in fucosylation, we examined the number of fucose residues in C/H-F and C/H-FS glycans. This analysis revealed that the that the normalized absolute peak intensity (NAPI) of mono-fucosylated glycans was little but statistically significantly higher in KO samples, whereas the NAPI of multi-fucosylated glycans (>2 fucose residues) was notably lower in KO than in WT samples (Fig. 7e). Taken together, these results indicate that the lack of GolpHCat alters protein glycosylation, specifically fucosylation, in the hippocampus.

To explore the changes in biological functions arising from the lack of GolpHCat, we conducted a proteomic analysis on membrane proteins from hippocampal brain samples of both WT and GolpHCat KO mice using LC-MS (Supplementary Fig. 8e). We found 29 differentially expressed proteins present only in WT and 53 differentially expressed proteins present only in GolpHCat KO mice (Supplementary Fig. 8f and Supplementary Table 3). Among these, the expression of certain ion channels, including inward rectifier potassium channel 4 (KCNJ4) and potassium voltage-gated channel subfamily C member 1 (KCNC1) increased, whereas the expression of sodium channel subunit beta-3 (SCN3B) decreased (Supplementary Fig. 8g, h and Supplementary Table 3). To examine the roles of these differentially expressed proteins, we analyzed their molecular functions using Gene Ontology (GO) analysis (Supplementary Fig. 8i). We found that "Transporter activity" was decreased in KO mice (Supplementary Fig. 8i). To predict the effect of these differentially expressed proteins on biological functions, we performed a protein network cluster analysis based on the results of the GO analysis (Supplementary Fig. 8j). The results predicted that these differentially expressed proteins play vital roles in learning and memory (Supplementary Fig. 8j). Taken together, these results indicate that the lack of GolpHCat alters glycosylation in the hippocampus, which is expected to alter biological functions, especially hippocampus-related learning and memory.

## GolpHCat KO impaired hippocampal-dependent memory

To test if the lack of GolpHCat causes altered hippocampus-related learning and memory, we performed detailed electrophysiological and behavioral analyses of the well-established hippocampal spatial memory circuit of CA3 → CA1 synapses. Unexpectedly, the intrinsic neuronal excitability of CA1 pyramidal neurons was not altered in GolpHCat KO mice compared to WT mice (Supplementary Fig. 9a–c), although the shape of the action potential was slightly altered in KO mice (Supplementary Fig. 9d–i). To examine synaptic transmission and plasticity, we measured extracellular field excitatory postsynaptic potentials (fEPSP) at the CA3-CA1 synapses of WT and GolpHCat KO mice (Fig. 8a). We then examined the basal synaptic transmission with increasing stimulus intensities at the Schaffer collaterals and found no difference between WT and GolpHCat KO mice (Fig. 8b). To examine the presynaptic release probability, we measured paired-pulse ratios (PPRs) with increasing interpulse intervals and consistently found no significant differences between WT and KO mice at any interpulse interval (Fig. 8c), indicating that the basal synaptic connectivity and transmission were unaltered in KO mice. Finally, to examine the potential role of GolpHCat in synaptic plasticity, we performed high-frequency stimulation (HFS)-induced long-term potentiation (LTP) and found that it was significantly impaired in GolpHCat KO mice (Fig. 8d, e). Taken together, these results indicate that the lack of GolpHCat alters hippocampal LTP while leaving basal synaptic transmission unchanged. Finally, to investigate whether the lack of GolpHCat affects hippocampus-dependent memory, we subjected WT and GolpHCat KO mice to hippocampus-dependent spatial memory-related behavioral tasks, such as novel place recognition (NPR) and contextual fear tests (Fig. 8f). GolpHCat KO mice showed a significant impairment of both contextual spatial memory in NPR (Fig. 8g, h) and contextual fear memory in the fear test (Fig. 8i), with no change in anxiety levels in the elevated plus maze test (Supplementary Fig. 9j–o). Taken together, these results indicate that GolpHCat is required for hippocampal spatial and contextual memory, which is consistent with predictions based on the glycomics and proteomics analyses (Fig. 7).

Because GolpHCat is expressed in both hippocampal astrocytes and neurons (Fig. 6a, d), we investigated the cell type that majorly contributes to memory impairment in GolpHCat KO mice. We performed cell-type specific gene silencing (KD) using GolpHCat-shRNA-carrying viruses, followed by behavioral tests (Fig. 8j–q and

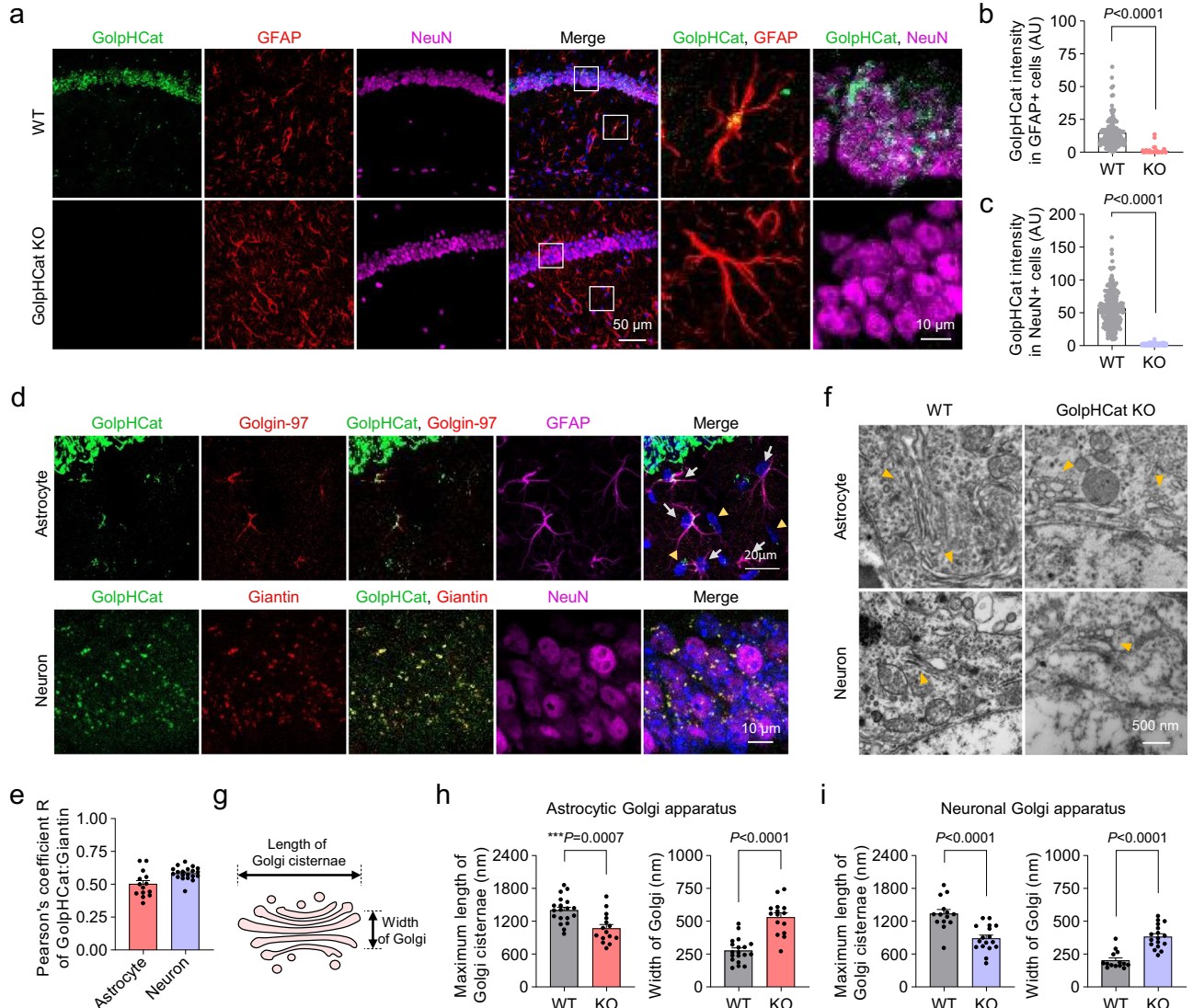

**Fig. 6 | Disruption of Golgi morphology in hippocampal astrocytes and neurons of GolpHCat KO mice. a** Immunostaining for GolpHCat, GFAP, and NeuN in the hippocampus of WT and GolpHCat KO mice (left). High-magnified images showing colocalization of GolpHCat with GFAP or NeuN (right). **b** Fluorescence intensity of GolpHCat immunoreactivities in GFAP+ cells of WT ($n = 206$ cells from four mice) and GolpHCat KO ($n = 102$ cells from three mice) mice. **c** Fluorescence intensity of GolpHCat immunoreactivities in NeuN+ cells of WT ($n = 241$ cells from four mice) and GolpHCat KO ($n = 259$ cells from three mice) mice. **d** Colocalization of GolpHCat with Golgin-97 or Giantin in hippocampal astrocyte (GFAP) and neuron (NeuN) of WT mice, respectively. **e** Pearson's correlation coefficient for colocalization of GolpHCat and Golgi markers in hippocampal astrocytes ($n = 14$ cells) and neurons ($n = 15$ cells) of WT mice. **f** TEM images of the Golgi apparatus in

hippocampal astrocytes and neurons of WT and GolpHCat KO mice. Yellow arrows indicate Golgi. **g** Diagram of Golgi structure for analysis used in (**h**, **i**). **h** Maximum length of Golgi cisternae (left) and width of Golgi (right) in hippocampal astrocytes of WT ($n = 19$ cells from three mice) and GolpHCat KO ($n = 15$ cells from three mice) mice. **i** Maximum length of Golgi cisternae (left) and width of Golgi (right) in hippocampal neurons of WT ($n = 14$ cells from three mice) and GolpHCat KO ($n = 17$ cells from three mice) mice. Data were presented as the mean ± SEM. Statistical analyses were performed using two-tailed Mann–Whitney test in **b** ($U = 365.5$), **c** ($U = 2$), **i** width ($U = 13$); two-tailed unpaired *t*-test in (**h**)-length ($t = 3.778$, df = 32), (**h**)-width ($t = 6.739$, df = 32), (**i**)-length ($t = 4.741$, df = 29). Source data and exact *p* values are provided as a Source Data file.

Supplementary Fig. 9p, q). A Cre-dependent shRNA expressing virus (Lenti-pSico-scrambled/GolpHCat shRNA-EGFP) and a cell type-specific Cre-expressing virus (AAV-GFAP-Cre-mCh for astrocyte; Fig. 8j, and Supplementary Fig. 9r or AAV-CaMKIIα-Cre-mCh for neuron; Fig. 8n, and Supplementary Fig. 9s) were co-injected in the hippocampal CA1 region. Both astrocytic and neuronal GolpHCat gene-silenced mice showed impaired spatial and contextual memory in the NPR test recall (Fig. 8k, l, o, p) and contextual fear test (Fig. 8m, q). Taken together, these results indicate that GolpHCat in both astrocytes and neurons is critical for spatial and contextual memory in the hippocampus.

## Discussion

In the present study, we provide unprecedented insights into the identification and function of TMEM87A as a bona fide cation channel within the Golgi apparatus. We also provide a comprehensive understanding of TMEM87A's structure-activity relationship by elucidating the ion conduction pathway and gating mechanism through the determination of two distinct cryo-EM structures bound to PE and gluconate. Contrary to a previous study that failed to observe single-channel currents in TMEM87A-reconstituted liposome patch recordings[22], our investigation employing different lipid compositions successfully demonstrates genuine voltage-dependent single-channel currents, providing unequivocal

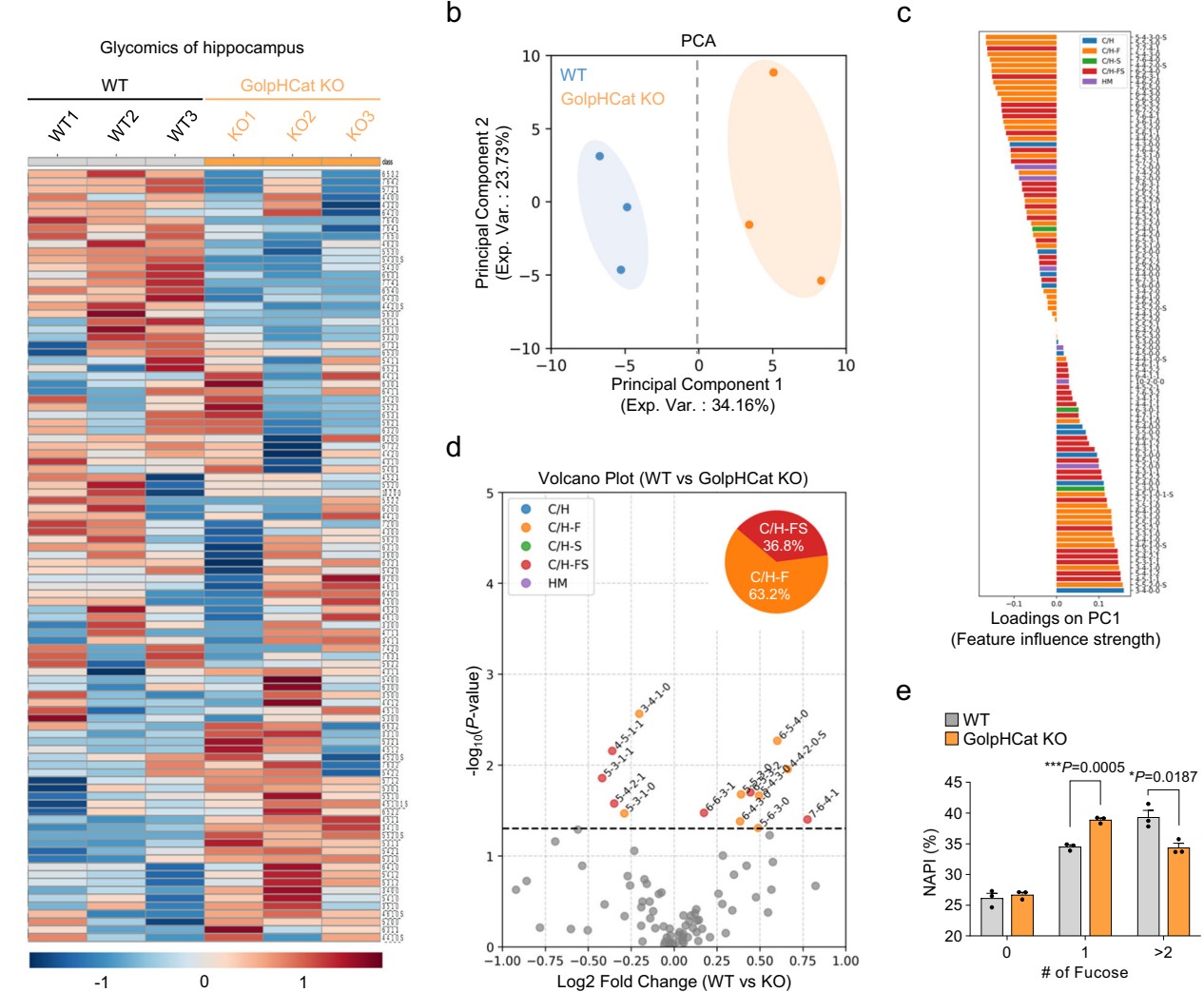

**Fig. 7 | A deficit of GolpHCat altered glycosylation in the hippocampus.**
**a–e** Comparison of 99 N-glycans found in the hippocampus of WT (*n* = 3 mice) and GolpHCat KO (*n* = 3 mice) mice. **a** Heat map for the hierarchical clustering of N-glycans. The scale bar indicates z-scores of standardized glycan value with higher or lower expressed glycans depicted in red or blue, respectively. (Sugar code: Hex_HexNAc_Fuc_NeuAc_(HexA)_sulfation(S)). **b** PCA analysis of hippocampus samples between WT and GolpHCat KO. **c** Feature influence strength for principal component 1 in (**b**). **d** Volcano plot. The red dots represent significant expression C/

H-F glycans, the orange dots represent significantly expression C/H-FS glycans, and the gray dots represent insignificant expressed glycans. **e** Comparative distribution of all fucosylated glycans NAPI by the number of fucose residues (*n* = 3 mice per group). Data were presented as the mean ± SEM. Statistical analyses were performed using two-tailed *t*-test in (**d**); two-tailed unpaired *t*-test in (**e**) (#0, *t* = 10.18, df = 4; #1, *t* = 0.6246, df = 4; #>2, *t* = 3.827, df = 4). Source data and exact *p* values are provided as a Source Data file.

evidence that TMEM87A is a bona fide ion channel. Our research surpasses previous structural studies by achieving higher resolution cryo-EM structures at 3.1 and 3.6 Å, in comparison to the previously published structure at 4.7 Å[22]. Based on this enhanced resolution, we are the first to identify the mysterious phospholipid as PE, while Hoel et al. only suggested a phospholipid-bound structure without specifying the type of phospholipid[22]. We also propose the PE-mediated voltage-dependent gating mechanism of TMEM87A, thereby shedding light on its intricate functionality. Moreover, we introduce gluconate as a potent blocker of TMEM87A and demonstrate its inhibitory effect on TMEM87A-mediated currents. Finally, we propose a critical function for TMEM87A in the brain and cognition, specifically highlighting its involvement in hippocampal spatial memory. In summary, our comprehensive structural and molecular studies significantly contribute to a deeper understanding of TMEM87A's role as a cation channel in the Golgi apparatus and the brain.

We have identified a unique voltage-dependent nonselective cation channel, which we named GolpHCat. According to the brain RNA-seq database, it is highly expressed in the brain[24,25]. Single-channel

analysis of GolpHCat demonstrates that, unlike any other known voltage-dependent channels, GolpHCat shows a skewed U-shaped voltage-dependent open probability curve, centered around 0 mV (Fig. 2c). Other voltage-gated channels typically exhibit a unidirectional sigmoidal activation curve[56], whereas GolpHCat displays a unique bidirectional activation curve at both negative and positive potentials. The one-and-only resembling channel is perhaps the voltage-dependent anion channel VDAC, which displays an upside-down U-shaped voltage-dependent open probability curve, centered around 0 mV[57,58]. Based on this unique voltage-activation property, one can make several interesting conjectures about how the Golgi membrane potential is maintained by GolpHCat; (1) GolpHCat's primary function might be to clamp the resting Golgi membrane potential at 0 mV by opening at both negative and positive offset voltages and resetting the voltage to near 0 mV (by depolarization and hyperpolarization upon channel opening, respectively), (2) the unique U-shaped voltage-dependent open probability curve should render the resting Golgi membrane potential set to 0 mV, independent of the

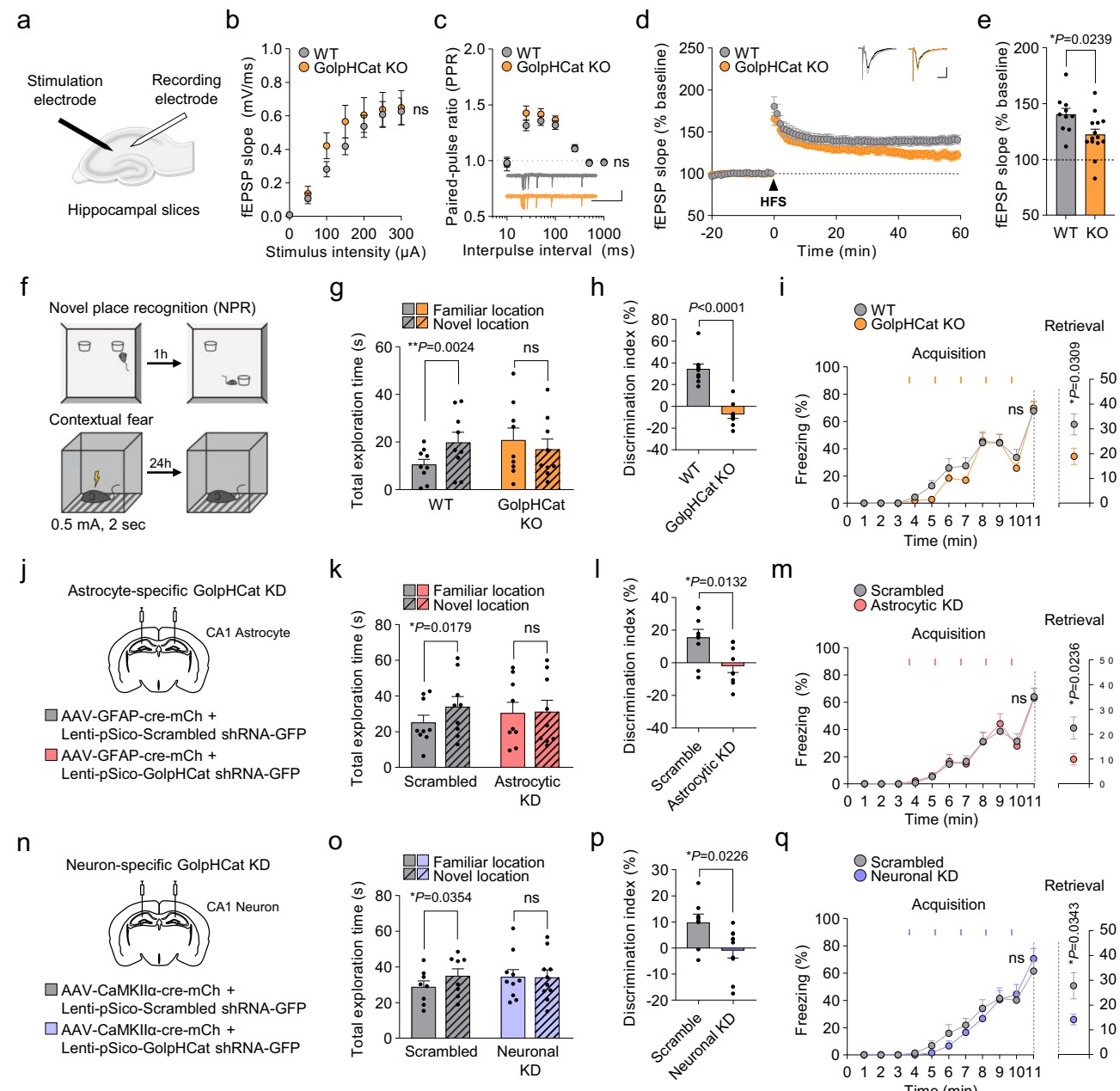

**Fig. 8 | GolpHCat contributes to hippocampal spatial and contextual memories. a** Schematic diagram of fEPSP recording in the Schaffer-collaterals pathway. **b** Input-output curve for fEPSP slope obtained with increasing stimulus intensities in WT and GolpHCat KO mice. **c** Paired-pulse ratios (PPR) obtained with increasing interpulse intervals. Inset: representative fEPSP traces. Scale bar: 1 mV and 500 ms. **d** HFS (1 s at 100 Hz)-induced LTP. Inset: representative fEPSP traces from before and after HFS-induced LTP. Scale bar: 0.5 mV and 10 ms. **e** Slope of fEPSP over the last 5 min. $n = 10$ cells from three mice for WT and $n = 14$ cells from three mice for GolpHCat KO in (**b**–**e**). **f** Schematic diagram of the NPR and contextual fear task. **g** Total object exploration time during the test phase of the NPR task. **h** Discrimination index during the test phase of the NPR task. **i** Percentage of freezing during the acquisition phase of contextual fear task (left) and retrieval phase during the 5 min (right). The orange rectangular indicates the time point of shock. $n = 9$ mice for WT and $n = 9$ mice for GolpHCat KO in (**g**–**i**). **j** Illustration of virus injection for Scrambled ($n = 9$) and astrocyte-specific GolpHCat KD (Astrocytic KD; $n = 9$) in the stratum radiatum of the hippocampal CA1 region. **k** Total object exploration time during the test phase of the NPR task. **l** Discrimination index during the test phase of the NPR task. **m** Percentage of freezing during the acquisition phase of contextual fear task (left). Percentage of freezing during the

retrieval phase during the 5 min (right). The pink rectangular indicates the time point of shock. $n = 9$ mice for Scrambled and $n = 9$ mice for Astrocytic KD in (**k**–**m**). **n** Illustration of virus injection for Scrambled and neuron-specific GolpHCat KD (Neuronal KD) in the pyramidal layer of the hippocampal CA1 region. **o** Total object exploration time during the test phase of the NPR task. **p** Discrimination index during the test phase of the NPR task. **q** Percentage of freezing during the acquisition phase of contextual fear task (left). Percentage of freezing during the retrieval phase during the 5 min (right). The purple rectangular indicates the time point of shock. $n = 8$ mice for Scrambled and $n = 10$ mice for Neuronal KD in (**o**–**q**). Data were presented as the mean ± SEM. Statistical analyses were performed using two-way ANOVA followed by Šídák's multiple comparisons test in **b** ($F_{(6,32)} = 0.8259$), **c** ($F_{(6,132)} = 1.021$), **i** acquisition, ($F_{(10,160)} = 0.7738$), **m** acquisition ($F_{(10,160)} = 0.2676$), **q** acquisition ($F_{(10,160)} = 1.002$); two-tailed unpaired $t$-test in **e** ($t = 2.426$, df = 22), **i** retrieval ($t = 2.367$, df = 16), **l** ($t = 2.786$, df = 16), **m** retrieval ($t = 2.502$, df = 16), and **p** ($t = 2.523$, df = 16); two-tailed paired $t$-test in **g** (WT ($t = 4.361$, df = 8), KO ($t = 2.275$, df = 8), **k** (Scrambled ($t = 2.969$, df = 8), Astrocytic KD ($t = 0.3143$, df = 8), **o** (Scrambled ($t = 2.600$, df = 7), Neuronal KD ($t = 0.1794$, df = 9); two-tailed Mann–Whitney test in (**h**) ($U = 0$), and **q** retrieval ($U = 16$). Source data and exact $p$ values are provided as a Source Data file.

concentration changes in luminal $Na^+$ and $K^+$ ions, and (3) GolpHCat might be able to achieve this voltage-clamping effect, regardless of the presence of any leak channels.

In this study, we determined the high-resolution cryo-EM structures of hGolpHCat complexed with the Golgi membrane phospholipid PE and the pharmacological inhibitor gluconate. While future studies are required to examine its function, our structural analysis of hGolpHCat, complemented by MD simulations and electrophysiological analyses, provides crucial insights into the ion conduction pathway of hGolpHCat as a nonselective voltage-dependent cation channel. Although residues on the funnel-shaped luminal vestibule can in principle attract cations, the R2-fatty acid chain of PE and key residues that interact with its head group appear to occlude the ion conduction pathway, indicating that hGolpHCat is impermeable to ions under cryo-EM conditions (resting state). Notably, the putative ion conduction pathway is lined with highly conserved hydrophobic and electropositive residues. Although currently unclear, we speculate that conformational changes in the voltage sensor TM3 in response to Golgi membrane depolarization under physiological conditions may lead to the opening of the ion conduction pathway in hGolpHCat (Fig. 5j). Further structural and functional studies are needed to elucidate the gating dynamics of GolpHCat.

We observed the GOLD domain in the ELD of TMEM87A, similar to transmembrane Emp24 domain-containing protein (TMED/p24), acyl-coenzyme A binding domain-containing protein 3 (ACBD3), and translocating chain-associating membrane protein (TRAM), in addition to Wntless. Many GOLD-containing protein families are well-known, and their functions have been studied. For example, TMED/p24 has been implicated in various cellular processes, including the biogenesis of coat protein vesicles[59] and has been suggested to act as a cargo receptor that facilitates transport along the secretory pathway[60,61]. The human Golgi-resident protein, ACBD3 plays a role in preserving the integrity of the Golgi structure through its interaction with giantin, which in turn affects the movement of proteins between the endoplasmic reticulum and Golgi apparatus[62]. A splice variant of Toll/interleukin-1 receptor/resistance protein domain-containing adapter molecule 2 (TICAM2) called TRAM with a GOLD domain is involved in signal transduction pathways, where it participates in intracellular signaling events related to Golgi functions[63,64]. Therefore, like other GOLD-containing proteins, GolpHCat may also play a role in several functions, such as trafficking, secretion, and sorting of yet unidentified proteins. Future work is needed to identify proteins interacting with the GOLD domain of TMEM87A using deeper proteomic analysis of mouse brain samples and to determine the structure of interacting protein complexes.

Our study proposes a counter-cation channel, GolpHCat, that facilitates proper luminal acidification of the Golgi apparatus for its normal morphology and functions. We generated GolpHCat KO mice and observed disrupted Golgi morphology, such as fragmentation and swelling, and altered protein glycosylation. It has been reported that a well-organized, ribbon-shaped Golgi structure is necessary for proper glycosylation in each cisterna of the Golgi apparatus[1,2,65]. When GolpHCat is removed, it is possible that the lack of GolpHCat aberrantly drives excessive $Cl^-$ and $H^+$ influx into the lumen through GPHR[3] and NHE7[11,12], respectively, thereby lowering the resting pH to 6.3. Simultaneously, excessive $Cl^-$ and $H^+$ influx creates an osmotic pressure, thereby allowing water molecules to flow in through the aquaporins (AQPs) and, owing to an osmotic pressure, causing the Golgi to swell. These effects may impair Golgi homeostasis, leading to altered protein glycosylation. Golgi pH homeostasis is also important for the distribution and activity of glycosylation enzymes. Elevated pH within the Golgi apparatus disrupts N-glycosylation by causing Golgi glycosyltransferases to be mislocalized[66,67]. Deviation of pH from its normal range can greatly alter the activity of enzymes within the Golgi, depending on their specific pH sensitivity[68,69]. In our study, it is

possible that the distribution and activities of fucosyltransferase enzymes were altered in GolpHCat KO mice, leading to the altered pattern of fucosylated glycans (Fig. 7a–e). We observed altered glycan patterns, with an increase in mono-fucosylated glycans and a decrease in multi-fucosylated glycans in the GolpHCat KO samples (Fig. 7e). This suggests that fucosyltransferase (FUT)8, associated with core fucosylation (1 fucose), may become more active, whereas other FUTs may be less active, due to Golgi pH perturbations. Future studies are needed to investigate whether FUT enzymes are altered in GolpHCat-KO mice. Taken together, our results propose that GolpHCat, as a counter-cation channel, is a key molecule for the maintenance of Golgi homeostasis and structure, allowing for normal Golgi functions, such as protein glycosylation, especially fucosylation, in the brain.

Our study raises an importance of Golgi pathology such as morphological and functional alteration in cognitive impairments. We firstly propose that not only neuronal but also astrocytic Golgi pathology, caused by the impaired Golgi pH, is largely responsible for cognitive impairment. Understanding the molecular mechanism of Golgi pathology is expected to shed light on therapeutic approaches for cognitive impairment found in various neurodegenerative diseases.

## Methods

### Ethics statement
All experimental procedures were approved by the Institutional Animal Care and Use Committees (IACUCs, #2018-IBS-20) of the Institute for Basic Science (IBS, Daejeon, South Korea).

### Animals
All mice were maintained under a 12:12-h light-dark cycle (lights on at 8:00 AM) and had ad libitum access to food and water. All mice were housed in groups of 4–5 per cage. Adult male C57BL/6 J wild-type mice (6–7 weeks old) were used for behavioral tests with or without virus injection. All experiments were done with sex- and age-matched controls.

### Generation of TMEM87A knockout mice
We requested the generation of a TMEM87A knockout mice line from GHBio (Daejeon, Korea). TMEM87A KO allele was generated by using the CRISPR/Cas9-mediated homologous recombination. Two guide RNAs (gRNAs), which bound to intron 9 or intron 10 and deleted the whole exon 10, including the GYG sequence, were designed with the following spacer sequences: spacer of gRNA1, 5′-TAAGCCAAGTACTAGCACGT-3′; spacer of gRNA2: 5′-GATGAAAGAAGTAGTGAGCT-3′. Protospacer adjacent motif (PAM), AGG, for the corresponding gRNA was located on the 3 bp downstream of each targeted DNA sequence. The targeting construct was injected into a C57BL/6 J mouse ES cell line with the two gRNAs. PCR and sequencing analysis were used to identify ES cell clones with proper targeting. Obtained female knockout mice were maintained by crossing with male WT mice. Genotypes were determined by PCR using the following primers; Forward: 5′-TGTGCACATAACTGAGGTCAT-3′; Reverse: 5′-GCTTCACTGCAATCTTTCTGC-3′.

### TMEM87A KO mice
GolpHCat KO mice (9–15 weeks) and WT littermates were used for IHC, multi-omics, slice patch, and behavior tests.

### CHO-K1 cells and human astrocytes
Chinese hamster ovary-K1 (CHO-K1, referred to as CHO) cells and human astrocytes were purchased from the Korean Cell Line Bank and abm (Richmond), respectively. CHO cells and human astrocytes were maintained in F-12 (Gibco), and DMEM (Corning) respectively. Both media were supplemented with 25 g/L glucose, 4 g/L L-glutamine, 1 g/L sodium pyruvate, 10% heat-inactivated fetal bovine serum (FBS, Gibco), and 1% penicillin-streptomycin (Gibco). Cells were incubated in a humidified 5% $CO_2$ incubator at 37°C.

## Sequence analysis

The partial sequence alignment of the selectivity filter of the potassium channels with hTMEM87A/B and the multiple sequence alignments of TMEM87A isoforms was performed using Clustal Omega (https://www.ebi.ac.uk/Tools/msa/clustalo/). Accession numbers are as follows: hTMEM87A isoform 1, NP_056312.2; isoform 2, NP_001103973.1; isoform 3, NP_001273416.1; hTMEM87B, NP_116213.1; Kv1.1, NP_000208.2; TWIK-1, NP_002236.1; TASK-1, NP_002237.1; TRAAK, NP_201567.1; THIK-1, NP_071337.2; HCN1, NP_066550.2.

Hydrophobicities of human and mouse TMEM87A were calculated in ProtScale (https://web.expasy.org/protscale/) using the Kyte&Doolittle, window size:21, 100%. The signal peptide in the N-terminal of TMEM87A and cleavage site position were predicted in the SignalP-5.0 (https://services.healthtech.dtu.dk/service.php?SignalP-5.0).

## Construct and shRNA cloning for *TMEM87A*

Open reading frame of human *TMEM87A*-transcript variant 1 (NM_015497; SC108269), 3 (NM_001286487; SC336449), and mouse *TMEM87A*-transcript variant 1 (NM_173734; MC201598) were purchased from Origene. The coding sequences of these genes were PCR amplified and then subcloned into CMV-pIRES2-DsRed/iRFP vector using the SalI/BamHI restriction enzyme sites or CMV-EGFP-N1 vector using the EcoRI/AgeI restriction enzyme sites using the cloning kit (EZ-Fusion™ HT Cloning core Kit, Enzynomics). Mutant forms, shRNA-insensitive form, and isoform 1 truncation of human *TMEM87A* were made using the mutagenesis kit (EZchange™ Site-directed Mutagenesis kit, Enzynomics). Information on oligomer sequences for cloning and mutagenesis were listed in Supplementary Table 4.

The pSicoR and pSico vectors were used for shRNA knockdown in vitro and in vivo, respectively. The targeted sequences for human TMEM87A and mouse TMEM87A were 5′-GGATTGGTGCTGT-CATCTTCC-3′ and 5′-GGATTGGTGCTGTCATCTTTC-3, respectively.

For making the human TMEM87A clone construction that is not sensitive to shRNA (shRNA-insensitive human TMEM87A), six nucleotides that make the same amino acid due to the redundancy of genetic code were changed in the human TMEM87A shRNA target site (shRNA-insensitive human TMEM87A sequence 5′- GGATTGGTGCTGT-CATCTTCC-3′ included six mismatches).

## Immunocytochemistry

Human astrocytes were cultured on 0.1 mg/ml poly-D-Lysine (PDL; P6407, Sigma-Aldrich)-coated coverslips in 24 wells for 24 h. Cells were fixed with 4% paraformaldehyde (PFA) in PBS for 20 min at room temperature before washing three times with PBS and incubating in blocking solution (2% donkey serum (Genetex, GTX27475), 2% goat serum (Abcam, ab7481) and 0.03% Triton-X100 (Sigma, 93443) for permeabilization in PBS) at room temperature (RT) for 1 h. For immunostaining, the samples were incubated overnight at 4 °C with primary antibodies: rabbit anti-TMEM87A (1:100, Novus Biologicals, NBP1-90531), mouse anti-Giantin (1:100, Abcam, ab37266), mouse anti-Golgin-97 (1:100, Invitrogen, A-21270) chicken anti-GFAP (1:200, Millipore, AB5541) diluted in blocking solution. After incubation overnight, the samples were washed three times with PBS and then incubated for 1 h at RT with secondary antibodies: anti-rabbit IgG Alexa Fluor 488 (1:500, Jackson lab), anti-mouse IgG Alexa Fluor 594 (1:500, Jackson lab), and anti-chicken IgG Alexa Fluor 647 (1:500, Jackson lab) diluted in the blocking solution. After secondary antibody incubation, samples were washed three times with PBS, and DAPI solution (1:2,000, Pierce, 62248) was added during the second washing step. Finally, samples were mounted with a fluorescent mounting medium (Dako, S3023). Fluorescence images were taken using a Zeiss LSM 900 confocal microscope with a 63X lens and all images are z-maximum projection. Quantification was done by ImageJ software. Representative figures were obtained from three independent experiments.

## Membrane localization of TMEM87A isoforms

Human astrocytes were cultured on PDL-coated coverslips for 24 h before transfection. Cells were transfected with TMEM87A shRNA-mCh with each shRNA-insensitive EGFP-tagged TMEM87A isoform for 48 h. Cells were washed once with PBS, then fixed with 4% paraformaldehyde in PBS for 20 min at RT and washed three times. DAPI solution was added during the second washing step. Finally, samples were mounted with a fluorescent mounting medium. Fluorescence images were taken at 405 nm for DAPI, 488 nm for EGFP, and 594 nm for mCh using a Zeiss LSM 900 confocal microscope with a 63X lens.

## Illumina Hiseq library preparation and sequencing

RNA was isolated from cultures of human astrocytes using a Qiagen RNEasy Kit (Cat. No #74106). Sample libraries were prepared using the Ultra RNA Library Prep kit (NEBNext, E7530), Multiplex Oligos for Illumina (NEBNext, E7335), and polyA mRNA magnetic isolation module (Invitrogen, Cat. No. #61011). Full details of the library preparation and sequencing protocol are provided on the website and previously described[70]. The Agilent Bioanalyser and associated High Sensitivity DNA Kit (Agilent Technologies) were used to determine the quality, concentration, and average fragment length of the libraries. The sample libraries were prepared for sequencing according to the HiSeq Reagent Kit Preparation Guide (Illumina, San Diego, CA, USA). Briefly, the libraries were combined and diluted to 2 nM, denatured using 0.1 N NaOH, diluted to 20 pM by addition of Illumina HT1 buffer, and loaded into the machine along with read 1, read 2, and index sequencing primers. After the 2 × 100 bp (225 cycles) Illumina HiSeq paired-end sequencing run, the data were base called, and reads with the same index barcode were collected and assigned to the corresponding sample on the instrument, which generated FASTQ files for analysis.

## NGS data analysis

BCL files obtained from Illumina HiSeq2500 were converted to FastQ and demultiplexed based on the index primer sequences. The data was imported to Partek Genomics Suite (Flow ver 10.0.21.0328; copyright 2009, Partek, St Louis, MO, USA), where the reads were further processed. Read quality was checked for the samples using FastQC. High-quality reads were aligned to the *Homo sapiens* (human) genome assembly GRCh37 (hg19, NCBI using STAR (2.7.8a). Aligned reads were quantified to the human genome assembly transcript model (hg19 - RefSeq transcripts 93) and normalized to obtain fragments per kilobase million (or FPKM) values of positively detected and quantified genes. Alternate splice variants for the genes were detected during quantification and also normalized to obtain FPKM values for the alternatively spliced variants.

## pH measurement with sensor imaging

For the pH measurement, pSicoR-Scrambled/TMEM87A shRNA-DsRed and B4GALT1-Ratiometric pHluorin2 were transiently co-transfected into human astrocytes one day before imaging. For the measurement of the Golgi pH buffer capacity, we obtained basal pH images and then treated 50 mM NH$_4$Cl. Fluorescence live images were excited at wavelengths of 405 and 475 nm, and the emitted fluorescence was captured through a spectral slit for wavelengths of 508 nm. The data were imported into Microsoft Excel for calculation of the intensity ratios (405 nm:475 nm) and further analyses. Golgi pH values were estimated using a pH calibration curve based on modified methodology from the previous study[71]. Briefly, calibration was performed with human astrocyte cells expressing GPI(plasma membrane targeting)-Ratiometric pHluorin2. The cells were first washed with PBS, followed by treatment with various extracellular solutions of different pH levels for 10 min, after which the fluorescence intensity was measured. This process is replicated with all calibration buffers to establish a standard curve.

## Whole-cell patch-clamp in CHO-K1 cells overexpressing hTMEM87A WT or mutants

Although endogenous human TMEM87A mainly localized to the Golgi membrane, overexpressed hTMEM87A is partially localized to the plasma membrane[21] (Supplementary Fig. 6f). Therefore, whole-cell patch clamping was performed to measure the channel activity of hTMEM87A localized at the plasma membrane.

For whole-cell patch-clamp recording, TMEM87A WT or mutants cloned into pIRES2-DsRed vectors (Addgene) were transiently transfected into CHO-K1 cells one day before patch recording. A transfection reagent (Lipofectamine 3000; Invitrogen, L3000001) was used for transfection in all experiments. After 24 h, cells were seeded onto 0.1 mg/ml PDL-coated coverslips and used for whole-cell patch-clamp recording within 12 h.

Unless otherwise indicated, the bath solution contained (in mM) 150 NaCl, 3 KCl, 10 HEPES, 5.5 glucose, 2 MgCl$_2$, and 2 CaCl$_2$ with pH adjusted to 7.3 by NaOH (320–325 mOsmol/kg). Borosilicate glass pipettes (Warner Instrument Corp., USA, GC150F-10) were pulled and had a resistance of 4–6 MΩ in the bath solution when filled with a pipette solution contained (in mM) 130 K-gluconate, 10 KCl, 10 HEPES, 10 1,2-Bis(2-aminophenoxy)ethane-$N$,$N$,$N'$,$N'$-tetraacetic acid (BAPTA) with pH adjusted to 7.3 by KOH (290 ~ 310 mOsmol/kg). Cells were held at −60 mV. For recording with voltage-clamp ramp protocol, currents were measured under the 1000-ms-duration voltage ramps descending from +100 to −150 mV with 10 s time intervals. For recording with voltage-clamp step protocol, currents were measured under the step pulses from +100 to −150 mV with 100 ms with 1 s time intervals. Electrical signals were amplified using MultiClamp 700B (Molecular Devices, USA). Data were acquired by Digitizer 1550B (Molecular Devices) and pClamp 11 software (Molecular Devices, USA) and filtered at 2 kHz. These machines were used in all electrophysiology experiments in this paper.

To determine the contribution of Na$^+$ in the bath solution for the inward currents, 150 NaCl-containing bath solution was replaced with 150 NMDG-Cl under the same other condition. For measuring the cation permeability of TMEM87A, 150 Na$^+$ in the bath solution was substituted with 150 X, where X was K$^+$ or Cs$^+$ (pH 7.3 was adjusted with KOH and CsOH, respectively). The pipette solution was prepared as described above. For calculating the relative permeability ratio of TMEM87A, we measured the differences in reversal potential between two cationic I-V curves. The permeability ratio, $P_X/P_{Na}$, was estimated using the modified Goldman–Hodgkin–Katz Eq. (1) as reported previously:

$$\Delta E_{rev} = \frac{RT}{F} * \ln\left(\frac{P_{Na}[Na^+]_{150\,Na} + P_X[X]_{150\,Na}}{P_{Na}[Na^+]_i + P_X[X]_i}\right) - \frac{RT}{F} * \ln\left(\frac{P_{Na}[Na^+]_{150\,X} + P_X[X]_{150\,X}}{P_{Cl}[Na^+]_i + P_X[X]_i}\right). \tag{1}$$

To determine the impermeability of anion, 150 NaCl containing bath solution was replaced to 150 Na-isethionate under the same other condition. The I-V curves in this experiment were corrected with the LJP (LJP: NaCl, +13 mV; Na-isethionate, −11.6 mV).

To assess the inhibitors as a TMEM87A blocker, they were contained in the bath solution and then recorded using the following concentration: Gadolinium chloride (GdCl$_3$) (0.3, 1, 3, 10, and 30 μM, Sigma), Na-gluconate (gluconate) (0.01, 0.03, 0.1, 1, and 10 μM, Sigma). The IC$_{50}$ values were estimated by non-linear regression analysis, using the following Eq. (2):

$$Y = \frac{100}{1 + \left(\frac{IC_{50}}{X}\right)^h} \tag{2}$$

where $X$ is the inhibitor concentration, $IC_{50}$ is the concentration required for half-maximal inhibition, and $h$ is the Hill slope constant ($h$: gadolinium, −1.498; gluconate, −1.301).

To measure the extracellular pH-dependency of TMEM87A currents, different pH solutions were made using the composition previously described, and the pH was manipulated with HCl or NaOH.

## Surface biotinylation assay and western blot

We selected HEK293T cells for our surface biotinylation experiments due to their superior transfection efficiency compared to CHO-K1 cells. The biotinylation assay was performed using the Pierce™ Cell Surface Protein Biotinylation and Isolation Kit (Thermo Scientific) as per manufacturer's instructions. In detail, hTMEM87A WT or its point mutants (E279A, E298A, D442A, Y237A, E272A, K273A, S301A, K304A, R305A, R309A, S415F, E288R, and G318AY319AG320A) cloned into the pIRES2-DsRed vector were transiently transfected into HEK293A cells (5 × 10$^6$ cells in 100 mm dish). After 40 h, cells were rinsed with PBS and then incubated with 0.25 mg/ml Ez-Link™-Sulfo-NHS-SS-Biotin, a membrane-impermeable reagent, in PBS for 10 min at room temperature. Biotinylated cells were washed with ice-cold TBS three times, and harvested cell pellets were resuspended in 250 μl lysis buffer containing a complete protease inhibitor cocktail (Roche). After centrifugation (15,000×$g$ for 5 min at 4 °C), the supernatant was collected, and protein concentration was measured using the Bradford protein assay. 800 μg of protein was incubated with 250 μl of NeutrAvidin agarose resin (Thermo Scientific) on a shaker for 30 min at room temperature. After three times washing using the washing buffer in the Pierce™ Cell Surface Protein Biotinylation and Isolation Kit, biotinylated proteins were eluted using 100 μl elution buffer containing 10 mM DTT, separated by 10 % SDS–PAGE and transferred to PVDF membranes. The membranes were blocked with 5% skim milk in TBST for 1 h at RT, and washed with TBST three times. Then, the membranes were incubated with primary antibodies: rabbit anti-TMEM87A antibodies (1:1000, Novus Biologicals) and mouse anti-β-actin antibodies (1:2000, Santa Cruz) overnight at 4 °C. The membranes were washed with TBST three times and incubated with the corresponding horseradish peroxidase-conjugated secondary antibodies [HRP-linked anti-rabbit IgG (Cell Signalling) and HRP-linked anti-mouse IgG, (Thermo Scientific) for 1 h at RT. After washing three times with TBST, immune-reactive protein bands were detected using EzWestLumiOne (ATTO).

## Cloning, expression, and purification of recombinant hTMEM87A

The cDNA encoding human TMEM87A (hTMEM87A, NP_056312.2, M1-E555) followed by a TEV protease cleavage sequence (ENLYFQG), a PreScission Protease cleavage sequence (LEVLFQGP), EGFP (M1-239K), a thrombin cleavage sequence (LVPRGS) and a Twin-strep-tag were cloned into the BamHI and XhoI sites of a pcDNA3.4 (Invitrogen). All hTMEM87A mutants were created by site-directed mutagenesis using the WT construct as a template. Constructs and primers are listed in Supplementary Table 5.

Recombinant hTMEM87A protein was transiently expressed in Expi293F cells (Thermo Fisher Scientific) according to the manufacturer's instructions. Briefly, 200 μg of plasmid DNA was transfected into 200 ml of Expi293F cells (3.0 × 10$^6$ cells/ml) using Expifectamine (Thermo Fisher Scientific). Cells were cultured in Expi293 expression medium (Thermo Fisher Scientific) at 37 °C and 8% CO$_2$ with shaking (orbital shaker, 120 rpm). After 20 h, the enhancer (Thermo Fisher Scientific) was supplemented to the culture, then further incubated for 30–34 h at 30 °C.

Cell pellets were resuspended in 20 ml HN buffer [50 mM HEPES pH 7.5, 250 mM NaCl, and 1x complete protease inhibitor cocktail (Roche)] and lysed by sonication (total 2 min, 1 s with intervals of 5 s, 20% amplitude). After ultracentrifugation (Beckman Ti70 rotor, 150,000×$g$ for 1 h), the collected membrane fraction was homogenized

with a glass Dounce homogenizer in 20 ml buffer [HN buffer + 1% (w/v) *n*-dodecyl β-ᴅ-maltoside (DDM; Anatrace) and 0.2% (w/v) cholesteryl hemisuccinate (CHS; Anatrace)] and solubilized for 2 h at 4 °C. The insoluble cell debris was removed by ultracentrifugation (Beckman Ti70 rotor, 150,000×g for 1 h), and the supernatant was incubated with 2 ml Strep-Tactin resin (IBA Lifesciences) for 30 min at 4 °C. After washing with 10 column volumes of wash buffer [HN buffer + 0.05% (w/v) DDM and 0.01% (w/v) CHS], hTMEM87A-EGFP-Twin strep tag was eluted with 5 ml elution buffer [HN buffer + 0.05% (w/v) DDM/CHS + 10 mM desthiobiotin]. After concentration using an Amicon Ultra centrifugal filter (100-kDa cut-off; Millipore), hTMEM87A-EGFP-Twin strep was further purified by size exclusion chromatography (SEC) using a Superose 6 Increase 10/300 GL column (Cytiva) equilibrated with a final buffer [HN buffer + 0.01% (w/v) DDM and 0.002% (w/v) CHS]. The peak fractions were immediately used for preparing proteoliposomes.

For structural studies, the incubated supernatant with 2 ml Strep-Tactin resin was washed with 10 column volumes of wash buffer [TN buffer (50 mM Tris pH 9.0, 250 mM NaCl, and 1x complete protease inhibitor cocktail) + 0.05% (w/v) lauryl maltose neopentyl glycol (LMNG; Anatrace) and 0.01% (w/v) CHS], hTMEM87A-EGFP-Twin strep tag was eluted with 5 ml elution buffer [TN buffer, 0.05% LMNG, 0.01% CHS, and 10 mM desthiobiotin]. After concentration using an Amicon Ultra centrifugal filter, hTMEM87A-EGFP-Twin strep was further purified by size exclusion chromatography (SEC) using a Superose 6 Increase 10/300 GL column equilibrated with a final buffer [TN buffer + 0.01% (w/v) LMNG and 0.002% (w/v) CHS]. The collected peak fractions were concentrated to -0.8 mg/ml using an Amicon Ultra centrifugal filter and immediately used for the cryo-EM grid preparation for the hTMEM87A structure. For the complex cryo-EM structure of hTMEM87A with gluconate (hTMEM87A-Gluc) and hTMEM87A A308M, HN buffer was used instead of Tris buffer during purification. Other protein solubilization and purification conditions were the same as those for hTMEM87A. Before freezing grids, 10 mM of sodium gluconate (Sigma-Aldrich) was added to purified hTMEM87A (0.7 mg/ml) and incubated for 1 h on ice.

### Reconstitution of human TMEM87A
A total of 10 mg of lipids (8:2, POPC:POPG; Avanti Polar Lipids) was dissolved in 1 mL chloroform in the glass tube, dried to a thin film under a nitrogen stream, and further dried overnight under a vacuum. A total of 2 mL dehydration/rehydration (D/R) buffer (5 mM HEPES, 200 mM KCl pH 7.2 adjusted by KOH) was added to the lipids, and the solution was vortexed for 60 s before being bath sonicated until transparent for 20 min. Purified hTMEM87A was added to 2 mg lipids in D/R buffer and reconstituted into liposomes at a 1:100 protein-to-lipid ratio. D/R buffer was used to bring the volume to 1 mL and roller mix at room temperature for 1 h. After the beads settled to the bottom of the tube to eliminate detergents, the supernatant was collected and ultracentrifuged for 45 min at 4 °C and 250,000×g. Pelleted proteoliposomes were resuspended in 80 μL D/R buffer by gently pipetting and used on the day. Three to four spots of 20 μL of proteoliposomes were placed on a glass coverslip coated with 0.1 mg/ml PDL. Proteoliposomes were vacuum-dried at room temperature for 6 h at 4 °C and then rehydrated with 20 μL DR buffer to each spot with wet filter paper overnight at 4 °C (8–24 h) for patch-clamp recording.

### Single-channel recording and analysis
A previously reported protocol[72] was used for the TMEM87A single-channel recordings in the liposome. All recordings were performed with the attached liposome patch. For the TMEM87A single-channel recording, the symmetric solution was used in bath and pipette solutions followed by: (in mM) 200 KCl, 5 HEPES with pH adjusted to 7.2 by KOH. Notably, for making the blister, 40 mM MgCl2 was added in bath solution 30 min before the patch and was maintained throughout the

recording. Borosilicate glass pipettes were pulled and polished to a resistance of 3–6 MΩ in the bath solution. The currents were recorded at RT and holding current as indicated in the data. The single-channel analysis was performed in Clampfit 10.7.

### Cryo-EM sample preparation and data collection
Quantifoil R 1.2/1.3 Cu 200-mesh holey carbon grids (SPI SUPPLIES) were glow-discharged for 75 s at 15 mA (PELCO easiGlow Glow Discharge Cleaning system, Ted Pella). Then, 4 μl of the purified hTMEM87A, hTMEM87A-Gluc, hTMEM87A A308M were applied to the grid at 100% humidity at 4 °C. After 7 s blotting, grids were plunged into liquid ethane using a FEI Vitrobot Mark IV (Thermo Fisher Scientific). Micrographs were acquired on a Titan Krios G4 TEM operated at 300 keV with a K3 direct electron detector (Gatan) at the Institute for Basic Science (IBS), using a lit width of 20 eV on a GIF-quantum energy filter. EPU software was used for automated data collection at a calibrated magnification of ×105,000 under the single-electron counting mode and correlated-double sampling (CDS) mode[73], yielding a pixel size of 0.849 Å/pixel. The micrograph was dose-fractionated to 57 frames under a dose rate of 7.95 e⁻/pixel/sec with a total exposure time of 6.14 s, resulting in a total dose of about 67.72 e⁻/Å². A total of 10,377 movies for hTMEM87A, 13,099 movies for hTMEM87A-Gluc, and 4565 movies for hTMEM87A A308M were collected with a nominal defocus range from −0.8 to −1.9 μm. Detailed parameters are summarized in Supplementary Table 2.

### Cryo-EM data processing, model building, and refinement
The detailed image processing workflow and statistics are summarized in Supplementary Fig. 3b–h, 3m–s, Supplementary Fig. 7, and Supplementary Table 2. Micrographs were subjected to patch motion correction and patch CTF estimation in cryoSPARC v.3.3.2[74]. For the hTMEM87A data set, 96,330 particles were first picked by a blob picker of cryoSPARC. Then, 2D class average images were generated as templates for subsequent reference-based auto-picking. A total of 8,101,104 particles from the complete datasets were binned four times, and to identify higher quality particles, subsequent 2D classification, Ab initio, and heterogeneous refinement were performed in cryoSPARC. The resulting 445,198 particles from the 3D classes showing good secondary structural features were re-extracted into the original pixel size for further 3D refinements. Non-uniform refinement[75] and CTF refinement[76] improved the particle alignment and map quality. The final refinement yielded a map at an overall -3.1 Å resolution according to the 0.143 cut-off criterion[77]. For hTMEM87A-Gluc, reference-based picked 6,035,205 particles were processed similarly to hTMEM87A data processing. The final non-uniform refinement from 201,915 particles yielded a map at an overall -3.6 Å resolution. For hTMEM87A A308M, reference-based picked 4,758,336 particles were processed similarly to hTMEM87A data processing. The final non-uniform refinement from 360,876 particles yielded a map at an overall -3.1 Å resolution. The mask-corrected Fourier shell correlation (FSC) curves were calculated in cryoSPARC, and reported resolutions were based on the gold-standard Fourier shell correlation (FSC) = 0.143 criteria. Local resolutions of density maps were estimated by Blocres[78]. Model building for hTMEM87A was initiated using the module 'Map to model' in PHENIX package[79] and a model generated by AlphaFold[32,80]. The model was then subjected to iterative manual and automated refinement rounds in PHENIX and Coot[81]. The final refinement statistics are summarized in Supplementary Table 2.

### Model analysis
A cavity search using a Solvent Extractor from the Voss Volume Voxelator server[82] was performed using an outer-probe radius of 5 Å and an inner-probe radius of 1.2 Å. The Dali server[29] was used to search protein structures having a similar fold. All molecular graphics figures were prepared with UCSF ChimeraX[83] and PyMOL[84].

## Gaussian accelerated molecular dynamics (GaMD) simulations

Gaussian accelerated molecular dynamics (GaMD) is an unconstrained enhanced sampling method that smooths the potential energy surface and reduces the energy barriers of biomolecular processes by adding a harmonic boost potential[37,38].

$$V^*\left(\vec{r}\right) = V\left(\vec{r}\right) + \Delta V\left(\vec{r}\right), \quad (3)$$

Where $V^*\left(\vec{r}\right)$ is the modified potential, $V\left(\vec{r}\right)$ is the system potential and $\Delta V\left(\vec{r}\right)$ is the harmonic boost potential. Along the simulation time, the harmonic boost potential is only added when system potential drops below reference energy:

$$\Delta V\left(\vec{r}\right) = \begin{cases} \frac{1}{2}k\left(E - V\left(\vec{r}\right)\right)^2, & V\left(\vec{r}\right) < E \\ 0, & V\left(\vec{r}\right) \geq E, \end{cases} \quad (4)$$

Where $E$ is the reference energy, and $k$ is the harmonic force constant. The two adjustable parameters $E$ and $k$ can be determined by applying the following criteria:

$$V_{max} \leq E \leq V_{min} + \frac{1}{k}, \quad (5)$$

Where $V_{max}$ and $V_{min}$ are the maximum and minimum potential energies, respectively.

The PE-bound hTMEM87A structure was used as a starting point and minimized in a two-stage geometry optimization approach using Gaussian accelerated molecular dynamics (GaMD) simulation. First, a short minimization of the water molecules positions, with positional restraints on the protein, ligand, and P31 atoms of the membrane, was performed with a force constant of 10 kcal/mol Å⁻² at constant volume periodic boundary conditions. Second, an unrestrained minimization including all atoms in the simulation cell was carried out. The minimized system was gently heated in two phases. First, the temperature was increased from 0 to 100 K in a 20 ps step. Harmonic restraints of 10 kcal/mol Å⁻² were applied to the protein, ligand, and membrane. Second, the temperature was slowly increased from 100 K to the production temperature (310.15 K) in a 100 ps step. In the second phase, harmonic restraints of 10 kcal/mol Å⁻² were applied to the protein, ligand, and P31 atoms of the membrane. The Langevin thermostat was used to control and equalize the temperature. The initial velocities were randomized in the heating step. In the heating and following steps, bonds involving hydrogen were constrained with the SHAKE algorithm, and the time step was set at 2 fs, allowing potential inhomogeneities to self-adjust. The equilibration step was performed in three stages. First, 5 ns of MD simulation under NVT ensemble and periodic boundary conditions were performed to relax the simulation temperature. Second, 5 ns of MD simulation under an NPT ensemble at a simulation pressure of 1.0 bar was performed to relax the density of the system. The semi-isotropic pressure scaling using the Monte Carlo barostat was selected to control the simulation pressure. Third, an additional 5 ns of MD simulation was performed to relax the system further. A cutoff value of 11 Å was applied to Lennard-Jones and electrostatic interactions.

After equilibration, an extra short 5 ns of MD simulation followed by 45 ns GaMD simulation (the boost potential is applied) was carried out in order to collect potential statics for calculating the acceleration parameters. Finally, a 500 ns of GaMD production run was performed. In total, we performed three independent simulations (i.e., 1.5 μs accumulated time). All GaMD simulations were performed using the AMBER2021 package[85] and applying the 'dual-boost' potential, where one boost potential is applied to the dihedral energetic term and the other to the total potential energetic term of the force field. The reference energy was set to the upper bound, which provides a more

aggressive boost. The upper limit of the boost potential standard deviation, σ0, was set to 6.0 kcal/mol.

## Molecular dynamics simulations and analysis for PE binding to hTMEM87A

All MD simulation systems were built from the cryo-EM structure of hTMEM87A (PDB ID: 8HSI) using Membrane Builder in CHARMM-GUI[86]. The missing loops (L148–K167 and S193–L202) were reconstructed using Modeller[87]. Using the PropKa program, the protonation state for K273 was determined as LYN, H187, and H403 as HID, and other histidine residues as HIE. All systems were solvated in ~150 mM KCl solution, and the periodic simulation boxes were about $10 \times 10 \times 14$ nm³ large. The Amber FF14SB, Lipid17, and TIP3P force fields were used for protein, lipid, and water, respectively[88,89]. The standard CHARMM-GUI equilibration protocol was followed to equilibrate the systems. The Particle-mesh Ewald method[90] was used for the electrostatic interaction, and a cut-off length of 0.9 nm for the van der Waals interaction. The production trajectories were integrated with a time step of 2 fs using OpenMM[91]. The temperature was kept at 310.15 K via the Langevin dynamics with a friction coefficient of 1 ps-1, and the pressure was retained at 1 bar via a Monte Carlo barostat with a coupling frequency of 5 ps-1. The trajectory analysis was performed with MDAnalysis[92], and the first 100 ns was discarded when calculating interaction energies. Supplementary Data 1–4 contain the initial and final structures of MD simulations.

From simulations of the lipid binding process (m->p), m-L + p* -> m + p-L, where m stands for membrane, we estimate the timescale for the hTMEM87A-PE binding process to be ~100 ns. We then try to estimate the timescale of the unbinding process by calculating the binding free energy $\Delta F_{m \to P}(L)$ of hTMEM87A-lipid binding process according to the linear interaction energy (LIE) model[42]. For that purpose, we simulated five 1 μs trajectories for the solvated system ($S_{p-L}$) of the cryo-EM structure embedded in a simple Golgi model membrane (m), which has a PC-to-PE ratio of 3:1, and one 1 μs trajectory for the solvated system ($S_{m-L}$) that consists of only the membrane. From the p-L trajectories, we computed average coulombic and van der Waals interaction energies corresponding to the L->p process of L(g) + p* -> p-L as

$$\Delta E_{L \to p}^{Q}(L) = \Delta E_{p-L}^{Q} - \Delta E_{P^*}^{Q} - \Delta E_{L(g)}^{Q} \quad (6)$$

$$\Delta E_{L \to p}^{vdW}(L) = \Delta E_{p-L}^{vdW} - \Delta E_{P^*}^{vdW} - \Delta E_{L(g)}^{vdW}. \quad (7)$$

From the m-L trajectory, we similarly computed for the L->m process of L(g) + m -> m-L.

$$\Delta E_{L \to m}^{Q}(L) = E_{m-L}^{Q} - E_{m^*}^{Q} - E_{L(g)}^{Q} \text{ and } \Delta E_{L \to m}^{vdW}(L) = E_{m-L}^{vdW} - E_{m^*}^{vdW} - E_{L(g)}^{vdW} \quad (8)$$

We can then obtain the average interaction energies for the m->p process by

$$\Delta E_{m \to p}^{Q}(L) = <\Delta E_{L \to p}^{Q}(L)>_{p-L} - <\Delta E_{L \to m}^{Q}(L)>_{m-L} \quad (9)$$

$$\Delta E_{m \to p}^{vdW}(L) = <\Delta E_{L \to p}^{vdW}(L)>_{p-L} - <\Delta E_{L \to m}^{vdW}(L)>_{m-L} \quad (10)$$

Finally, the binding free energy is estimated by the LIE formula of

$$\Delta F_{m \to P}(L) = 0.5 <E_{m \to p}^{Q}(L)> + 0.16 <\Delta E_{m \to p}^{vdW}(L)> \quad (11)$$

## Immunohistochemistry

Adult mice were deeply anesthetized with isoflurane and transcardially perfused with saline, followed by cold 4% paraformaldehyde in 0.1 M PBS. Brains were post-fixed in 4% paraformaldehyde for 24 h at 4 °C and 30% sucrose for 48 h at 4 °C. Frozen brains in OCT embedding compound solution were cut into 30 μm coronal sections. Sectioned brains were washed three times in PBS and incubated in a blocking solution (2% donkey serum, 2% goat serum, and 0.3% Triton-X100 for permeabilization in PBS) at room temperature (RT) for 1 h. Samples were incubated in the blocking solution (2% donkey serum, 2% goat serum, and 0.3% Triton-X100 for permeabilization in PBS) at RT for 1 h. For immunostaining, the samples were incubated overnight at 4 °C with primary antibodies: rabbit anti-TMEM87A (1:100, Novus Biologicals, NBP1-90531), mouse anti-Giantin (1:100, Abcam, ab37266), chicken anti-GFAP (1:200, Millipore, AB5541), guineapig anti-NeuN (1:200, Millipore, ABN90) diluted in blocking solution. After incubation overnight, the samples were washed three times with PBS and then incubated for 1 h at RT with secondary antibodies: anti-rabbit IgG Alexa Fluor 488 (1:500, Jackson lab), anti-mouse/chicken IgG Alexa Fluor 594 (1:500, Jackson lab), and anti-chicken/guineapig IgG Alexa Fluor 647 (1:500, Jackson lab) diluted in blocking solution. After secondary antibody incubation, samples were washed three times with PBS, and DAPI solution was added during the second washing step. Finally, samples were mounted with a fluorescent mounting medium. Fluorescence images were taken using a Zeiss LSM900 confocal microscope with a 20X and 63X lens and obtained Z-stack images in 1–2 μm steps and the fluorescent intensity was analyzed using the Image J software. For colocalization analysis, fluorescence images were taken using the Zeiss Elyra 7 Lattice SIM with 63X lens and Pearson's correlation coefficient R was analyzed using the colocalization tools in ZEN Blue software.

## Transmission electron microscopy and analysis for Golgi morphology

Brains were fixed with 2% glutaraldehyde and 2% PFA in 0.1 M PBS for 12 h at 4 °C and washed in 0.1 M PBS, and then post-fixed with 1% OsO4 in 0.1 M PBS for 1.5 h. The samples were then dehydrated with increasing concentrations of ethanol (50–100%), infiltrated with propylene oxide for 10 min, embedded with a Poly/Bed 812 kit (Polysciences, USA), and polymerized for 18 h at 60 °C. The samples were sectioned into 200 nm with a diamond knife in the ultramicrotome (EM-UCT, Leica, USA) and stained with toluidine blue for observation with an optical microscope. Thin sections (70 nm) were double-stained with 5% uranyl acetate for 10 min and 1% lead citrate for 5 min. Images were taken using the transmission electron microscope (JEM-1011, JEOL, Japan) at the acceleration voltage of 80 kV and photographed with a digital CCD camera (Megaview III). Golgi morphologies were analyzed using the ZEN Blue software.

## Slice patch recording for hippocampal neurons and astrocytes

To test the intrinsic properties of the hippocampal neurons and astrocytes, slice recording was performed using a modified protocol from the previous reports[93]. Briefly, the brain was excised from the skull and sectioned in an ice-cold, oxygenated (95% O2/5% CO2) sucrose-based dissection buffer containing (in mM) 212.5 sucrose, 5 KCl, 1.23 NaH2PO4, 26 NaHCO3, 10 glucose, 0.5 CaCl2, 10 MgSO4, pH 7.4. Brain slices were transversely cut into 300 μm thick sections containing hippocampus using a vibrating microtome (DSK Linearslicer™ Pro7, DSK, Japan). Prepared brain slices were recovered and recorded in oxygenated (95% O2/5% CO2) artificial cerebrospinal fluid (aCSF) containing (in mM) 124 NaCl, 5 KCl, 1.25 NaH2PO, 26 NaHCO3, 2.5 CaCl2, 1.5 MgCl2, and 10 glucose for at least 1 h at 28 ± 1° prior to recording. Brain slices for the astrocyte patch were co-loaded with 0.5 μM SR-101 (Sigma; S7635) dye to identify the location of astrocytes in the CA1 region.

For rheobase and action potential recording in hippocampal pyramidal neurons, a patch electrode (6–8 MΩ) was filled with an internal solution (in mM): 145 K-gluconate, 10 HEPES, 5 KCl, 0.2 EGTA, 5 Mg-ATP, and 0.5 Na2-GTP, pH adjusted to 7.3, and osmolarity 295 mOsmol/kg). Measurement was performed in a whole-cell current-clamp configuration, with no membrane potential adjustment. Rheobase and action potential were measured by giving 5 or 20 pA depolarizing steps for 1 s injection with 3 s between steps, respectively. Frequency, spike half-width, spike rise, and spike decay values of Rheobase were analyzed by Mini analysis software (Synaptosoft). For passive conductance recording in hippocampal astrocytes, measurement was performed in a whole-cell voltage-clamp configuration, and we used the holding potential of −80 mV. Patch electrode (6–8 MΩ) was filled with an internal solution (in mM): 140 KCl, 10 HEPES, 5 EGTA, 2 Mg-ATP, 0.2 NaGTP, adjusted to pH 7.4 with KOH. Currents were measured under the 1000-ms-duration voltage ramps descending from +100 to −150 mV with 10 s time intervals. Passive conductance data analysis was performed using Clampfit (Molecular Devices)

## Field excitatory postsynaptic potential (fEPSP) recording

To test basal synaptic transmission, paired-pulse ratio (PPR), and long-term potentiation (LTP), brain slice preparation and fEPSP experiments were performed as described previously[93]. Briefly, the mouse was anesthetized with isoflurane and decapitated. Isolated brain from decapitation was cut into 400-μm-thick transverse hippocampal slices using a vibrating microtome (DSK) in ice-cold, oxygenated (95% O2/5% CO2) sucrose-based dissection buffer containing 5 KCl, 1.23 NaH2PO4, 26 NaHCO3, 10 glucose, 0.5 CaCl2, 10 MgSO4, and 212.5 sucrose (in mM).

Brain slices were recovered in oxygenated aCSF containing 124 NaCl, 5 KCl, 1.25 NaH2PO4, 2.5 CaCl2, 1.5 MgCl2, 26 NaHCO3, and 10 glucose (in mM) at 28 ± 1 °C for at least 1 h and subjected to the fEPSP recordings. To evoke fEPSP from the Schaffer collateral pathway, an electrical stimulus was delivered with a concentric bipolar electrode (CBBPE75, FHC, Bowdoin, ME, USA).

To record fEPSP in the Schaffer collateral pathway, an aCSF-filled recording pipette, fabricated from a borosilicate glass capillary (1–3 MΩ, Harvard Apparatus, USA), was placed in stratum radiatum of hippocampal CA1. The slope of fEPSP was acquired and analyzed with WinLTP v2.01 software (WinLTP Ltd., The University of Bristol, UK). For the basal synaptic transmission, stimulus intensity was increased by 50 pA from 0 to 300 pA. In subsequent experiments, the stimulus intensity was set to 40–45% of the maximum response. For the PPR experiment, two pulses were delivered at intervals of 10, 25, 50, 100, 250, 500, and 1000 ms, and the ratio was calculated by dividing the fEPSP slope from the second response by the one from the first response. For the LTP experiment, the slope of fEPSPs was monitored at 0.067 Hz (one pulse per 15 s) during the experiment. After obtaining a stable fEPSP response for at least 20 min, LTP was induced with a single high-frequency stimulation (HFS) (1 s at 100 Hz). To quantify the degree of potentiation, the fEPSP slopes over the last 5 min from each slice were averaged.

## Extraction of membrane protein from brain tissue samples

Brain tissue samples (WT, $n = 3$; GolpHCat KO, $n = 3$) were homogenized with a buffer consisting of 0.25 M sucrose, 20 mM HEPES-KOH pH 7.4, and a 1:100 protease inhibitor mixture by sonication. The protein concentration of the homogenized samples was determined using a Qubit 2.0 Fluorometer, and 250 μg of protein was used for membrane extraction. The lysates containing 250 μg protein were pelleted by ultracentrifugation at 200,000×g for 45 min in a homogenization buffer, then resuspended in 0.2 M Na2CO3 (pH 11), and centrifuged at 200,000×g for 45 min. Finally, the membrane fraction was extracted by once more resuspending 0.2 M Na2CO3 and centrifuging at 200,000×g for 45 min[94].

## Enzymatic release and enrichment of N-glycans

Each membrane fraction resolubilized in 50 μL deionized (DI) water was mixed with an equal volume of 200 mM $NH_4HCO_3$ and 10 mM dithiothreitol (Sigma-Aldrich) and then denatured for the thermal cycle (100 °C, 2 min). After cooling, 2 μL of peptide N-glycosidase F (New England Biolabs) was added, and the entire mixture was incubated in a water bath at 37 °C for 16 h. The mixture was then chilled in 80% (v/v) ethanol (Merck) at – 42 °C for 1 h, and the glycan-rich supernatant was collected by centrifugating and precipitating out the deglycosylated proteins. The supernatant fraction was vacuum-dried and followed by purifying the released N-glycans using porous graphitized carbon-solid phase extraction (PGC-SPE). Briefly, PGC cartridges (Agilent Technologies, USA) were washed with 6 mL of DI water and 6 mL of 80% acetonitrile (ACN) and 0.1% trifluoroacetic acid (TFA) in DI water (v/v). The cartridge was conditioned with 6 mL of DI water, followed by loading an aqueous N-glycan solution onto the cartridge. Continuously, N-glycans were eluted stepwise with 6 mL of 10% ACN (v/v), 20% ACN (v/v), and 40% ACN and 0.05% TFA (v/v) in DI water after washing with 8 mL of DI water. All fractions were vacuum-dried and resolubilized with 15 μL of DI water before LC/MS analysis.

## N-glycan profiling by nano-LC/Q-TOF MS

Purified glycans were analyzed at nano-LC/Q-TOF MS with a nano-LC chip consisting of a porous graphitized carbon analytical column (5 μm, 0.075 × 43 mm i.d.). Glycans were separated at 0.3 μL/min with a 65 min gradient using Buffer A (3.0% ACN with 0.1% formic acid (v/v) in water and Buffer B (90% ACN with 0.1% formic acid (v/v) in water). The LC gradient used was as follows: 2.5 min, 0% B; 20 min, 16% B; 30 min, 44% B; 35 min, 100% B; 45 min, 100% B; 45.01 min, 0% B. MS spectra were acquired in positive ionization mode with a mass range of m/z 500–2000 and an acquisition time of 0.63 s per spectrum.

After data acquisition, raw LC/MS data were processed by the Molecular Feature Extractor algorithm of MassHunter Qualitative Analysis software B.07.00 (Agilent Technologies). A list of all N-glycans was extracted using the previously optimized application of the spatial mouse brain glycome database[55]. N-glycan compositions were identified with a mass error tolerance of 10 ppm using computerized algorithms.

## Protein enrichment from mouse brain tissue

The dried membrane fraction was mixed with 100 μL of 50 mM ammonium bicarbonate. About 2 μL of 550 mM dithiothreitol was added, and samples were incubated in a water bath at 60 °C for 50 min. Chilled samples mixed with 4 μL of 450 mM indole-3-acetic acid (IAA) were incubated at room temperature for 45 min in the dark. The mixture was then digested with 10 μL of 0.2 g/L trypsin in a water bath at 37 °C for 16 h. Digested proteins were purified by C18 SPE. C18 cartridges (Thermo Fisher Scientific) were first washed with 6 mL of 0.1% TFA in DI water (v/v) followed by 6 mL of 80% ACN, and 0.1% TFA in DI water (v/v). The cartridge was conditioned with 6 mL of 0.1% TFA in DI water (v/v), followed by loading an aqueous digested protein solution onto the cartridge. Peptides were continuously eluted with 6 mL of 80% ACN, and 0.1% TFA in DI water (v/v) after washing with 6 mL of 0.1% TFA in DI water (v/v). The fractions were dried under vacuum and resolubilized with 100 μL of 0.1% TFA in DI water (v/v) before LC-MS/MS analysis.

## Proteome analysis with nano-LC/Orbitrap MS/MS

Proteins and glycoproteins were analyzed using a Thermo Scientific Ultimate 3000 RSLCnano system coupled to a Thermo Scientific Q Exactive Plus hybrid quadrupole Orbitrap mass spectrometer with a nano-electrospray ion source. Mobile phases A and B were water with 0.1% formic acid (v/v) and ACN with 0.1% formic acid (v/v), respectively. The samples were separated at a 0.3 μL /min flow rate for 130 min using PepMap RSLC C18 column (Thermo Scientific, 2.0 μm, 75 μm × 50 cm). The Orbitrap MS parameters were set as follows: survey scan of peptide precursors was performed at 70 K FWHM resolution in the range of m/z 350–1900. HCD fragmentation was performed on 27 at a resolving power setting of 17.5 K. Resulting fragments detected in the range of m/z 200–2000 were used.

The raw data of proteins were processed using MaxQuant v2.0.1.0. Data were searched against the UniProt/SwissProt mouse (Mus musculus) protein database with 17,127 total entries and contaminant proteins. Searches were performed with the following parameter: Data were filtered with a peptide-to-spectrum match (PSM) of 0.01 false discovery rates (FDR), 7 in minimum peptide length, 1 in minimum unique peptide, modification including oxidation and acetylation in protein N-terminal, and true of iBAQ and match between runs. LFQ intensity data was statistically calculated concentrations of proteins using Perseus. Gene Ontology (GO) term enrichment was performed using PANTHER, and network clusters were by Cytoscape using ClueGO and CluePedia.

## Novel place recognition behavioral task

A novel place recognition task was performed as previously described[95]. Animal moving in real time was recorded with EthoVision XT software (Noldus). For the experiment, mice were placed in an open field with two identical objects and given 10 min to explore these objects and returned to their home cage. After 1 h, mice were placed back into the open field with two identical objects with one object relocated to a novel place in the field (corner directly opposite to the object's previous location). Mice were recorded while exposed to this condition for 10 min to observe their spatial recognition memory. The Discrimination index was calculated as the percentage of time spent examining the object in the novel place over the total time spent examining both objects.

## Contextual fear behavioral task

On the training day, mice were placed in a standard fear conditioning shock chamber. Mice were allowed to explore the chamber freely for 5 min and then received the first electrical foot shock (0.5 mA, 2 s duration, followed by four more shocks at 1 min and 30 s intervals. After 24 h, on the test day, mice were placed back into the same chamber for 5 min to assess freezing during the retrieval phase. The mice's movements in the fear conditioning chamber were recorded using a near-infrared camera and analyzed in real-time with EthoVision XT software (Noldus). Freezing behavior was defined as immobility for more than 2 s. The freezing percentage was calculated as the immobile time divided by the total time.

## Stereotaxic virus injection into the hippocampal CA1

All viruses used in this study were produced at the Institute for Basic Science virus facility (IBS virus facility). Mice were placed in stereotaxic frames after being anesthetized with vaporized isoflurane (Kopf). The scalp was incised, and a hole was drilled into the skull above the CA1 (anterior/posterior, −1.5 mm; medial/ lateral, ±1.5 mm from bregma, dorsal/ventral, −1.8–2.0 mm from the brain surface). The virus was loaded into a glass needle and injected bilaterally into the CA1 at a rate of 0.1 μl/min for 5 min (total 0.5 μl) using a syringe pump (KD Scientific). In each experiment, AAV-GFAP-Cre-mCh, AAV-CaMKIIa-Cre-mch, Lenti-psico-Scramble-GFP, and Lenti-pSico-TMEM87A shRNA-GFP viruses were used. Three weeks after the virus injection, mice were used for behavioral experiments and Immunohistochemistry.

## Statistics and reproducibility

All experiments in this study are performed with at least three biological replicates (mice or cell culture experiments). Representative image data for Figs. 1a, 6a, d, f were also obtained from at least three biological replicates. All data were presented as the mean ± SEM and significant symbol and value were represented in each figure, legend of

figure, and source data, respectively. GraphPad Prism 9.4.1 software was used for statistical analysis. Normal distribution was first assessed using the D'Agostino-Pearson omnibus normality test for all experiments. Parametric tests (Student's two-tailed paired or unpaired $t$-test, one-way ANOVA) were used for data following a normal distribution. Non-parametric tests (Mann–Whitney test, Kruskal–Wallis test) were used for data not following a normal distribution. Samples that passed the normality test, but not equal variance test was assessed with Welch's correction. Two-way ANOVA followed by Šídák's post hoc test was used. The significance was represented as asterisks (*$p < 0.05$, **$p < 0.01$, ***$p < 0.001$, ****$p < 0.0001$, and ns non-significant).

### Reporting summary

Further information on research design is available in the Nature Portfolio Reporting Summary linked to this article.

## Data availability

The data that support this study are available from the corresponding authors upon request. The atomic coordinates have been deposited to the Protein Data Bank (PDB) under the accession numbers 8HSI (hTMEM87A); 8HTT (hTMEM87A with gluconate); and 8KB4 (hTMEM87A A308M). The cryo-EM maps have been deposited in the Electron Microscopy Data Bank under accession codes EMD-34998 (hTMEM87A); EMD-350178 (hTMEM87A with gluconate); and EMD-37069 (hTMEM87A A308M). The accession codes for the PDB structures are human TMEM87a, PDB: 8CTJ, ChRmine, PDB: 7W9W, Wntless, PDB: 7DRT, Glucagon receptor, PDB: 5YQZ, and TRIC-B1, PDB: 5EGI. Raw files obtained in the Next Generation RNA Sequencing experiments are available on NCBI GEO (Accession number GSE228084). The source data underlying all figures are provided as a Source Data file. Source data are provided with this paper.

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

## Acknowledgements

We are grateful to GHBio, South Korea for generating TMEM87A KO mice; Match Finder, South Korea, for TEM imaging. We are grateful to the staff of the Research Solution Center at IBS for help with cryo-EM data collection. Computational work for this research was performed on the data analysis hub (Olaf) in the IBS Research Solution Center. This work was supported by grants from the Institute for Basic Science (IBS-R030-C1 to H.M.K. and IBS-R001-D2 to C.J.L.). This work was also supported by Young Scientist Fellowship (IBS-R001-Y1) to W.K. from the Institute for Basic Science, the Bio & Medical Technology Development Program (NRF-2022M3E5F3080873), the Medical Research Center (MRC) grant (NRF-2018R1A5A2025286), and the Brain Pool Program (NRF- 2021H1D3A2A02038434, NRF-2021H1D3A2A02081370) funded by the Ministry of Science and ICT (MSIT) through the National Research Foundation of Korea (NRF) to S.C. We also thank the Korea Institute of Science and Technology Information (KISTI) Supercomputing Center (KSC-2021-CRE-0469).

## Author contributions

H.K. and A.-r.H. contributed equally and can be listed first in bibliographic documents. H.K., A.-r.H., H.M.K., and C.J.L. designed the experiments and analyzed the data; H.K. performed whole-cell patch recording, ICC, IHC, and pH imaging. H.K. and J.M.L. performed proteoliposome patch recording. A.-r.H. purified proteins and determined the cryo-EM structure; A.Z., M.A.M.-S., and S.C. designed, performed MD simulation and analyzed the data; J.Y. and H.K. performed surface biotinylation assay and western blot. H.K., W.K., and W.S.R. performed slice patch recordings. H.K. and M.B. conducted RNA sequencing. H.K. and H.L. conducted behavioral tests. H.J., H.Y.J., and H.J.A. performed proteomics and glycomics. J.K. analyzed glycomics data. H.K., A.-r.H., A.Z., S.C., H.M.K., and C.J.L. wrote the manuscript.

## Competing interests

The authors declare no competing interests.

## Additional information

¹Center for Cognition and Sociality, Life Science Cluster, Institute for Basic Science (IBS), 55 Expo-ro, Yuseong-gu, Daejeon 34126, Republic of Korea. ²IBS School, University of Science and Technology (UST), 217 Gajeong-ro, Yuseong-gu, Daejeon 34113, Republic of Korea. ³Center for Biomolecular and Cellular Structure, Life Science Cluster, Institute for Basic Science (IBS), 55 Expo-ro, Yuseong-gu, Daejeon 34126, Republic of Korea. ⁴Global AI Drug Discovery Center, College of Pharmacy and Graduate School of Pharmaceutical Science, Ewha Womans University, Seoul 03760, Republic of Korea. ⁵Graduate School of Analytical Science and Technology, Chungnam National University, Daejeon 34134, Korea. ⁶Department of Biological Sciences, Korea Advanced Institute of Science and Technology (KAIST), Daejeon 34141, Republic of Korea. ⁷These authors contributed equally: Hyunji Kang, Ah-reum Han, Aihua Zhang. ✉e-mail: sunchoi@ewha.ac.kr; hm_kim@kaist.ac.kr; cjl@ibs.re.kr

