## [Peer Review File · Nature Communications]

GolpHCat (TMEM87A), a unique voltage-dependent cation channel in Golgi apparatus, contributes to Golgi-pH maintenance and hippocampus-dependent memory

Editorial Note: Parts of this Peer Review File have been redacted as indicated to remove third party material where no permission to publish were obtainedEditorial Note: This manuscript has been previously reviewed at another journal that is not operating a transparent peer review scheme. This document only contains reviewer comments and rebuttal letters for versions considered at *Nature Communications*.

Reviewer #1 (Remarks to the Author):

In the revised version, Kang et al. have essentially addressed all comments and questions previously raised by the reviewers. Importantly, they have solved an additional structure of a GolpHCat variant, plus they have shortened some parts of the results and discussion sections, which were a bit exaggerated in the previous manuscript version.

I still have to state that this manuscript contains an impressive amount of work and data and really covers a topic from a molecular view (structure), via activity to an physiological impact (here learning/memory).

While I feel that the article can be an important contribution for the field, I still have a few suggestions which may help to further improve the article.

- Most importantly: The authors claim that a PE is bound, yet, I was surprised not to see a more direct evidence, such as a TLC or MS analysis. This would really prove that it is a PE.
- P5, second paragraph starting with "TMEM87A in proteoliposomes": Here the authors describe an U-shaped voltage-dependent activation curve when measured in proteoliposomes but not in cells. Previously, the problem of a random orientation of the channel in the proteoliposomal membrane was already mentioned and the authors addressed this in the response letter. Yet, the problem/situation is not clarified at all. Can the U-shaped activation curve observed in proteoliposomes not simply originate from the random orientation of the proteins? This would also explain why we do not observe such an activation curve with intact cells, where the protein likely does have a defined topology. This needs to be clarified.
- Entire article: At several places the text refers to wrong figures or figure panels. Please check. E.g. on page 6 the text refers to "Extended Data Fig. 5f". There is no figure panel 5f! On page 10 the text discusses the mito morphology and refers to Ext. Data Fig. 8c. This Fig. does not show any mito data.
- Entire article: The authors found differences with p values >0.01. While often p-values up to 0.05 are discussed as "significant", I encourage the authors to go through the article and may write something like "little but statistically significant" or so.
- The language is still odd at several places. E.g. on page 29 it says: "Protein localization with TMEM87A isoforms". Probably "membrane localization" "of" the isoforms is meant here?
- Figures: The authors present a huge amount of data and tried to place as much as possible into one figure. E.g. Fig 1 has 19 items! As a result, many figure panels are very small and essentially unreadable on a printed version. If the journal allows, I really encourage the authors to prepare more figures, which would allow enlarging individual panels.
- P 3, first part: It may be helpful to add "isoform 1" to some parts of the description, as this is not necessarily clear for the reader.
- P 3, 3rd paragraph: please check whether the given figures are correct. E.g. it says "extended Data Fig. 1h" and probably 1c and g are meant.
- P 3, last sentence: this was addressed already in the last review and answered by the authors: it still remains mystic while we find a protein that should be Golgi-localized, in the plasma membrane. I feel that this needs further explanation, and a link to a previous article is not enough.
- Page 7, second paragraph, middle: Here the D442A mutant is described and it is stated that the "mutants showed no change compared to WT". This is clearly visible in Figure 4d, yet the curves shown in Ext. Data Fig. 6b indicate a clear difference.
- P 9, second paragraph: The E288R mutant "affected hTMEM87A ion conductance". This is absolutely correct. Yet, the conductance rate was increased, and this was probably not really expected, or was it. The authors should briefly discuss this finding.

Reviewer #2 (Remarks to the Author):

The paper delineates the structure and functions of TMEM87A, also known as GolpHCat. The authors purified the protein and conducted CryoEM analysis, revealing that TMEM87A adopts a monomeric structure with an ion-conducting pathway. Consequently, the authors concluded that TMEM87A

functions as a cation channel, supported by evidence demonstrating its sensitivity to voltage steps in proteoliposomes. Additionally, the study highlights the significance of TMEM87A in the Golgi apparatus function of astrocytes and neurons, as knockout mice exhibited impaired learning and memory functions. The authors addressed the reviewer's questions by revising the manuscript. Their explanation is compelling, particularly considering that the present study achieved a higher resolution than Hoel's CryoEM analysis, allowing for a clearer identification of the ion conduction pathway. Another primary question posed by the reviewer pertained to the role of PE in voltage-induced channel openings. The authors clarified that technical limitations in manipulating the lipid environment within the plasma membrane prevented the removal or addition of PE to induce changes in channel openings. In response, they opted to mutate a residue in the pore region and observed that this mutation effectively blocked the currents. These responses are both reasonable and contribute to the overall convincing context of the present study. The revised manuscript meets the standards for acceptance and is suitable for publication in Nature Communications.

Reviewer #3 (Remarks to the Author):

The revised version of the manuscript entitled "GolpHCat (TMEM87A), a unique voltage-dependent cation channel in Golgi apparatus, contributes to Golgi-pH maintenance and hippocampus-dependent memory" by Kang, Han et al. describes the characterization of the subcellular localization, the structure and functions of TMEM87A. The authors provide evidence that the Golgi-localized TMEM87A which they renamed GolpHCat is a non-selective cation channel, regulating Golgi pH homeostasis. Diverse experimental methods have been employed to carry out this study, such as measurement of intracellular pH and ionic currents in cultured cells, biochemistry, cryo-EM for structural biology and study of hippocampal-dependent behavior in mice.

1- The authors have modified the Introduction to highlight that Hoel et al 2022 already published the Golgi-localization and the structure of TMEM87A. The revised version of the manuscript is thus improved regarding this point.

2- Page 27 of the rebuttal letter/ page 3 of the revised manuscript:

To my opinion, the use of the term "Golgi signal sequence" is inappropriate. The term "signal sequence" refers to a protein sequence which will induce the translocation of the protein in the endoplasmic reticulum. Once the protein enters the ER then it will be transported through the secretory pathway. Then other motifs or protein sequences are involved in defining its localization in intracellular compartments, such as KDEL motif for retrieval to the ER or transmembrane domains properties or motifs deciphering its localization in the Golgi apparatus.

In consequence, the authors must remove the term of Golgi signal sequence but rather use "Golgi-targeting motif" or equivalent.

In addition, the pictures shown in Extended Data Fig. 1e (and in the rebuttal letter page 9) do not convincingly demonstrate that the lack of this sequence abolishes the Golgi-localization of TMEM87A. Indeed, TMEM87A lacking its signal sequence is expected to be cytoplasmic (no translocation in the ER and further transport) and/or to be degraded.

On the last panel of Extended Data Fig. 1 e (high exposure time provided in the rebuttal letter p9), a cytoplasmic fluorescence signal is visible for the condition "sh-insensitive-TMEM87A-iso 3-EGFP" but as no non-transfected cells is included in the field, we cannot rule out that this signal corresponds to bleed-through of the mCherry signal. If the authors do not detect fluorescence of for the Delta-ss or isoform 3, they might also provide experiments performed in presence of MG132 to assess degradation of the construct.

Alternatively, is there any issue in the plasmid preventing efficient expression ?

3- Figure 1e: please also provide a picture with lower exposure time to detect Golgi-localization of overexpressed TMEM87A in CHO-K1/

In addition, please modify the title of Fig1 as measurement of currents have been performed in CHO-K1 cells (not in astrocytes), if I am not mistaken.

4- Page 30 of the rebuttal letter: the additional information provided by the authors regarding the surface biotinylation answers appropriately to my comments.

5- Fig 6d (rebuttal letter pages 33-34):

The co-localization of TMEM87A with the Golgi apparatus is still not convincing. In astrocytes the signal for Golgin97 is very diffuse. In addition, if I am not mistaken, visualization of Golgin-97 has been performed thanks to immunostaining. Consequently, I do not understand while only the cell in the middle of the field shows golgin-97 -positive signal. Golgin-97 is expected to be expressed also in the neighboring cell. The authors might consider fluorescence bleed-through of the GFAP signal to the channel of the Golgi-97 staining. To my opinion, the upper panel of Fig.6 d cannot be published as it is.

6- Minor point: page 6 of the revised manuscript

Is the reference to Extended Data Fig.5f the right one ?

The authors appropriately modified the revised version to take into account the other concerns that I raised after reviewing the initial version of the manuscript.

Authors' response to the reviewers' comments
"GolpHCat (TMEM87A), a unique voltage-dependent cation channel in the Golgi apparatus, contributes to Golgi-pH maintenance and hippocampus-dependent memory"

by Kang and Han et al.

NCOMMS-23-61122A

REVIEWER COMMENTS

Reviewer #1 (Remarks to the Author):

In the revised version, Kang et al. have essentially addressed all comments and questions previously raised by the reviewers. Importantly, they have solved an additional structure of a GolpHCat variant, plus they have shortened some parts of the results and discussion sections, which were a bit exaggerated in the previous manuscript version.

I still have to state that this manuscript contains an impressive amount of work and data and really covers a topic from a molecular view (structure), via activity to an physiological impact (here learning/memory).

While I feel that the article can be an important contribution for the field, I still have a few suggestions which may help to further improve the article.

We are grateful for Reviewer #1's positive evaluation of our work and deeply appreciate valuable suggestions, which have greatly contributed to improving our manuscript. We have carefully considered the suggestions provided and believe they will enhance the quality and impact of our article. We will address these suggestions in detail in the revised manuscript to ensure that our work continues to make a significant contribution to the field. Once again, we thank Reviewer #1 for insightful comments and constructive feedback.

• *Most importantly: The authors claim that a PE is bound, yet, I was surprised not to see a more direct evidence, such as a TLC or MS analysis. This would really prove that it is a PE.*

We appreciate the feedback provided by Reviewer #1 regarding the validation of our phosphatidylethanolamine (PE) assignment. As Reviewer #1's comments indicate, we conducted preliminary experiments aiming to utilize LC-MS analysis to directly identify bound PE from the purified protein. We followed methodologies outlined in previous publications (Shin et al., Nat Struct Mol Biol. (2024), PMID:38332368 and Schmidpeter, P. A. M. et al. Nat Struct Mol Biol. (2022), PMID:36352139), yet we encountered challenges in achieving optimal conditions for sample preparation, highlighting the necessity for further optimization and validation to ensure reliable results.

Despite our continued efforts, identifying the optimal conditions is expected to pose difficulties for the following reasons:

1. **The low abundance of bound PE:** Given that we assigned only one PE molecule per hTMEM87A, the amount is quite minimal. Consequently, not only is a substantial quantity of purified protein required for a single LC-MS experiment, but we also need to identify the optimal lipid extraction method for efficient PE extraction.

2. The complexity of lipid-protein interaction: Membrane proteins interact with a multitude of different lipids, rather than just one. In fact, in the our cryo-EM structure of hTMEM87A, various phospholipids were observed in the detergent micelle region. Unlike the protein region, the local resolution of micelle is relatively low, and the density is very weak. Therefore, we refrained from assigning any lipid therein. Detecting PE alone within such a lipid complex with reliability is likely to be challenging.

We hope that Reviewer #1 will recognize and understand the experimental limitations. We plan to conduct LC-MS analysis for future studies to reinforce our conclusions regarding PE assignment.

Our study utilized the high-resolution cryo-EM structure of hTMEM87A at 3.1 Å resolution (within the transmembrane domain reaches ~2.8Å), which provided robust evidence for the assignment of PE based on precise lipid density (Supplementary Fig. 3h and 3k). Additionally, MD simulations supported the preference for PE over phosphatidylcholine (PC) (Supplementary Fig. 6h-l). Although we acknowledge the potential value of employing techniques MS and TLC in future studies, we believe that our combined approach using cryo-EM and MD simulations offers convincing evidence for the assignment of PE, taking into account the inherent limitations of the alternative methods in this specific context.

We appreciate the valuable suggestion for further validation of the PE assignment using a broader range of methodologies in future studies, which would help strengthen our conclusions.

We have revised the manuscript (page 8 highlighted in blue) as follows:

"However, future studies employing techniques like mass spectrometry could further strengthen our conclusions and provide complementary evidence for PE in TMD."

• *P5, second paragraph starting with "TMEM87A in proteoliposomes": Here the authors describe an U-shaped voltage-dependent activation curve when measured in proteoliposomes but not in cells. Previously, the problem of a random orientation of the channel in the proteoliposomal membrane was already mentioned and the authors addressed this in the response letter. Yet, the problem/situation is not clarified at all. Can the U-shaped activation curve observe in proteoliposomes not simply originate from the random orientation of the proteins? This would also explain why we do not observe such an activation curve with intact cells, where the protein likely does have a defined topology. This needs to be clarified.*

We thank Reviewer #1's for critical comment. Firstly, we apologize for any confusion caused by our use of the term 'U-shaped activation curve' in the discussion, which may have led to misunderstanding. To clarify this, we have revised the term to 'U-shaped open probability curve' as highlighted blue on page 12.

Regarding the concern about the potential influence of random protein insertion on the observed U-shaped open probability curve in proteoliposomes, if the U-shaped open probability were solely a result of random protein insertion, a symmetrical U-shaped open probability curve would be expected. However, our data in Figure 2c show a skewed (non-symmetric) U-shaped open probability curve, indicating that the TMEM87A proteins have a preferential orientation within the liposome.

It is important to note that the skewed U-shaped open probability was specifically measured in

proteoliposome single-channel patch only and cannot be directly measured in whole-cell currents. However, the unitary current \times probability-voltage relationship of TMEM87A in proteoliposome patch (Figure 2e) bears similarity to inwardly rectifying currents observed in whole-cell currents (Figure 1f).

Reviewer #1's inquiry has prompted us to clarify these key points, and we appreciate the opportunity to provide further explanation.

- *Entire article: At several places the text refers to wrong figures or figure panels. Please check. E.g. on page 6 the text refers to "Extended Data Fig. 5f". There is not figure panel 5f! On page 10 the discuss the mito morphology and refer to Ext. Data Fig. 8c. This Fig. does not show any mito data.*

We sincerely apologize for any inconvenience caused and appreciate Reviewer #1's comment regarding the errors within the manuscript. In response to this valuable feedback, we have fixed the errors as follows:

"On page 6, "Extended Data Fig. 5f" has been corrected to "Supplementary Fig. 4f"

"On page 10, "Extended Data Fig. 8c" has been corrected to "Supplementary Fig. 8d".

- *Entire article: The authors found differences with p values >0.01. While often p- values up to 0.05 are discussed as "significant", I encourage the authors to go through the article and may write something like "little but statistically significant" or so.*

We appreciate Reviewer #1's comment regarding the point. we have made changes as follows:

"On page 3, "significantly lower" has been revised to "little but statistically significantly lower"

"On page 11, "significantly" has been revised to "little but statistically significantly"

- *The language is still odd at several places. E.g. on page 29 it says: "Protein localization with TMEM87A isoforms". Probably "membrane localization" "of" the isoforms is meant here?*

We thank Reviewer #1's comment regarding the language used in the manuscript. In response, we have made the following adjustments:

"Protein localization with TMEM87A isoforms" has been revised to "Membrane localization of TMEM87A isoforms"

- *Figures: The authors present a huge amount of data and tried to place as much as possible into one figure. E.g. Fig 1 has 19 items! As a result, many figure panels are very small and essentially unreadable on a printed version. If the journal allows, I really encourage the authors to prepare more figures, which would allow enlarging individual panels.*

We apologize for any inconvenience caused by the readability of our figures. In response, we have enlarged the figures and increased font size from 7 to 8 to improve readability.

If our manuscript is accepted for publication and the journal permits, we are committed to working with the editor to further address this issue. This may involve increasing the number of figures, moving some panels to supplementary figures, or employing other solutions to improve the clarity and readability of our data.

- *P 3, first part: It may be helpful to add “isoform 1” to some parts of the description, as this is not necessarily clear for the reader.*

We appreciate Reviewer #1's for constructive feedback. We have revised the manuscript to address the concerns raised. In response to Reviewer #1's comment, we have revised the manuscript (p.3 highlighted in blue), as follows:

“ We first analyzed the protein sequence of full-length TMEM87A and found that TMEM87A contains a GYG sequence, which is a signature selectivity filter of classical K⁺ channels²³ (Supplementary Fig. 1a), raising the possibility that full-length TMEM87A may be a cation channel. Full-length TMEM87A encodes a 63kDa protein with a predicted N-terminal Golgi signal sequence and seven transmembrane (TM) domains (Supplementary Fig. 1b, c). In humans, TMEM87A encodes three isoforms: isoform 1 is full-length with a predicted Golgi signal sequence and TMs, isoform 2 has no TM, and isoform 3 has no predicted Golgi signal sequence (Supplementary Fig. 1d). According to the brain RNA-seq database, full-length TMEM87A (isoform1) is highly expressed in both, neurons and astrocytes^{24,25}. Thus, based on bioinformatics analysis, TMEM87A is a potential candidate for the Golgi-resident cation channel in the brain.”

- *P 3, 3rd paragraph: please check whether the given figures are correct. E.g. it say “extended Data Fig. 1h” and probably 1c and g are meant.*

We appreciate Reviewer #1's comment and would like to confirm that the figures referenced in the manuscript are indeed correct as presented in our version. We apologize for any confusion that may have arisen.

To provide clarification, both Fig. 1c and Fig. 1d are related with Supplementary Fig. 1i (Supplementary Fig.1h in the previous manuscript version), with Fig. 1c corresponding to the top panel and Fig. 1d corresponding to the bottom panel.

As part of the adjustment, we have revised the sentence as follows:

“We found that gene silencing of TMEM87A by shRNA led to a more basic resting Golgi pH than non-silenced (scrambled) conditions (Fig. 1c and Supplementary Fig. 1i (top)). Furthermore, Golgi pH buffering capacity, as measured by the change in pH upon 50 mM NH₄Cl application, was little but statistically significantly lower in TMEM87A shRNA-transfected cells (Fig. 1d and Supplementary Fig. 1i (bottom)), indicating that TMEM87A contributes to Golgi pH buffering capacity.”

- *P 3, last sentence: this was addressed already in the last review and answered by the authors: it still remains mystic while we find a protein that should be Golgi-localized, in the plasma membrane. I feel that this need further explanation, and a link to a previous article is not enough.*

We appreciate Reviewer #1's constructive comments regarding the observed cell surface expression of TMEM87A in the heterologous overexpression system, as previously discussed in our first rebuttal.

We understand the importance of investigating the trafficking mechanism in overexpression system

responsible for the localization of TMEM87A in the plasma membrane, as raised by the reviewer. While we did not delve into this issue extensively in the current manuscript, we acknowledge that it remains a critical aspect requiring further elucidation.

In response to this concern, we reference a study by Gee, H. Y et al. published in *Cell* in 2011 (146(5), 746-760), which explores the concept of unconventional protein secretion triggered by ER-to-Golgi block or ER stress. This mechanism involves Golgi-independent protein trafficking pathways. We hypothesize that in our heterologous overexpression system, the overexpressed TMEM87A may induce an ER-to-Golgi block, consequently affecting the unconventional trafficking route and leading to the observed cell surface expression.

[Redacted]

Gee, H. Y et al. study published in Cell 146.5 (2011): 746-760

While we did not delve into this hypothesis in detail in the current manuscript, we are committed to exploring it further in future studies. Specifically, we plan to investigate the trafficking mechanisms involved in TMEM87A localization to the plasma membrane and its functional implications. We believe that a deeper understanding of this aspect will provide valuable insights into the physiological role of TMEM87A in cellular processes.

• *Page 7, second paragraph, middle: Here the D442A mutant is described and it is stated that the “mutants showed no change compared to WT”. This is clearly visible in Figure 4d, yet the curves shown in Ext. Data Fig. 6b indicate a clear difference.*

We sincerely appreciate Reviewer #1's valuable suggestion. In our analysis, we examined the channel activity of GolpHCat and its mutants using measured current densities [I (pA/pF)] at -150mA. Specifically, for D442A, we conducted whole-cell patch clamping experiments 10 times, selecting the measurement with the average value of I (pA/pF) at -150mA as the representative I-V curve. (Refer to Supplementary Fig. 6b).

Subsequently, in response to Reviewer #1's comment, we thoroughly reviewed all measured data for D442A. However, we did not observe any significant difference in current compared to the wild

type (WT) on average.

We believe this error occurred due to the unintentional selection of an I-V curve for D442A. We sincerely apologize for this oversight in the data analysis process. To address this issue, we have replaced the I-V curve for D442A with one that aligns more closely with the results presented in Fig. 4d. (see revised Supplementary Fig. 6b).

Supplementary Fig.6 | a, Representative current-voltage (I-V) curves of hTMEM87A WT. **b**, Representative I-V relationship of hTMEM87A mutants (E279A, E298A, and D442A) for NLV.

• P 9, second paragraph: The E288R mutant “affected hTMEM87A ion conductance”. This is absolutely correct. Yet, the conductance rate was increased, and this was probably not really expected, or was it. The authors should briefly discuss this finding.

We appreciate Reviewer #1’s suggestion regarding the increase in channel activity with the E288R mutation in hTMEM87A. As mentioned, E288 forms interactions with nearby basic residues (R39, K41, and H43), potentially stabilizing a specific domain orientation of hTMEM87A (as seen in Supplementary Fig. 4d, Patch 4). We initially hypothesized that disrupting these interactions through the E288R mutation introduce conformational change of TM3, potentially influencing a channel activity. Indeed, the mutation showed increased channel activity (Fig. 5i). E288R might cause conformational changes of TM3 or alter electrostatic interaction within TMD, possibly leading to more open and ion-permissive environment.

As Reviewer #1’s suggestion, we have revised manuscript (p.9 highlighted in orange) as follows:

“Moreover, we found that mutations in the loop residues flanking the TM3 helix (E288R on ELL1) resulted in increased channel activity than WT (Fig. 5i and Supplementary Fig. 6e). As noted, E288 interacts with nearby basic residues, maintaining the structural integrity of hTMEM87A (Supplementary Fig. 4d, Patch4). Replacing the negatively charged glutamate with positively charged arginine disrupts interactions with the basic residues, potentially introducing the conformational changes to TM3 and influencing channel activity. Alternatively, the mutation might alter electrostatic interactions within the TMD, leading to an environment more conducive to ion flow. To unravel the precise mechanism, further investigations are needed.”

Reviewer #2 (Remarks to the Author):

The paper delineates the structure and functions of TMEM87A, also known as GolpCat. The authors purified the protein and conducted CryoEM analysis, revealing that TMEM87A adopts a monomeric structure with an ion liposomes. Additionally, the study highlights the significance of TMEM87A in the Golgi apparatus function of astrocytes and neurons, as knockout mice exhibited impaired learning and memory functions. The authors addressed the reviewer's questions by revising the manuscript. Their explanation is compelling, particularly considering that the present study achieved a higher resolution than Hoel's CryoEM analysis, allowing for a clearer identification of the ion conduction pathway.

Another primary question posed by the reviewer pertained to the role of PE in voltage-induced channel openings. The authors clarified that technical limitations in manipulating the lipid environment within the plasma membrane prevented the removal or addition of PE to induce changes in channel openings. In response, they opted to mutate -conducting pathway. Consequently, the authors concluded that TMEM87A functions as a cation channel, supported by evidence demonstrating its sensitivity to voltage steps in proteo a residue in the pore region and observed that this mutation effectively blocked the currents. These responses are both reasonable and contribute to the overall convincing context of the present study.

The revised manuscript meets the standards for acceptance and is suitable for publication in Nature Communications.

We thank Reviewer #2 for their impressive summary and positive evaluation of our work. We are sincerely thankful for Reviewer #2's time, expertise, and dedication to advancing our manuscript.

Reviewer #3 (Remarks to the Author):

The revised version of the manuscript entitled "GolpHCat (TMEM87A), a unique voltage-dependent cation channel in Golgi apparatus, contributes to Golgi-pH maintenance and hippocampus-dependent memory" by Kang, Han et al. describes the characterization of the subcellular localization, the structure and functions of TMEM87A. The authors provide evidence that the Golgi-localized TMEM87A which they renamed GolpHCat is a non-selective cation channel, regulating Golgi pH homeostasis. Diverse experimental methods have been employed to carry out this study, such as measurement of intracellular pH and ionic currents in cultured cells, biochemistry, cryo-EM for structural biology and study of hippocampal-dependent behavior in mice.

1- The authors have modified the Introduction to highlight that Hoel et al 2022 already published the Golgi-localization and the structure of TMEM87A. The revised version of the manuscript is thus improved regarding this point.

We appreciate Reviewer #3's positive assessment of introduction part in revised manuscript.

2- Page 27 of the rebuttal letter/ page 3 of the revised manuscript:

To my opinion, the use of the term "Golgi signal sequence" is inappropriate. The term "signal sequence" refers to a protein sequence which will induce the translocation of the protein in the endoplasmic reticulum. Once the protein enters the ER then it will be transported through the

secretory pathway. Then other motifs or protein sequences are involved in defining its localization in intracellular compartments, such as KDEL motif for retrieval to the ER or transmembrane domains properties or motifs deciphering its localization in the Golgi apparatus. In consequence, the authors must remove the term of Golgi signal sequence but rather use “Golgi-targeting motif” or equivalent.

We appreciate Reviewer #3's valid comment regarding the use of the term 'Golgi signal sequence'. We agree that the term 'Golgi signal sequence' may not accurately reflect the function of the sequence in directing protein localization within the Golgi apparatus. In response to Reviewer #3's suggestion, we have revised the manuscript to use the term 'Golgi-targeting motif' instead, which more appropriately describes the function of the sequence.

Furthermore, we have updated the abbreviation from Δ ss (ss: signal sequence) to Δ , and this change has been reflected in the figures as well.

Thank you for bringing this to our attention, and we apologize for any confusion caused by the previous terminology. We believe that these adjustments enhance the clarity and accuracy of our manuscript.

In addition, the pictures shown in Extended Data Fig. 1e (and in the rebuttal letter page 9) do not convincingly demonstrate that the lack of this sequence abolishes the Golgi-localization of TMEM87A. Indeed, TMEM87A lacking its signal sequence is expected to be cytoplasmic (no translocation in the ER and further transport) and/or to be degraded.

On the last panel of Extended Data Fig.1 e (high exposure time provided in the rebuttal letter p9), a cytoplasmic fluorescence signal is visible for the condition “sh-insensitive-TMEM87A-iso 3-EGFP” but as no non-transfected cells is included in the field, we cannot rule out that this signal corresponds to bleed-through of the mCherry signal. If the authors do not detect fluorescence of for the Delta-ss or isoform 3, they might also provide experiments performed in presence of MG132 to assess degradation of the construct.

Alternatively, is there any issue in the plasmid preventing efficient expression ?

We apologize for the incomplete response regarding Extended Data Fig. 1e in our initial rebuttal and appreciate the opportunity to address this issue more thoroughly. To investigate the possibility of protein degradation, we performed both western blotting and ICC staining under various conditions, as recommended by Reviewer #3.

In response to Reviewer #3's suggestion, we performed ICC staining using TMEM87A-iso1/iso1 Δ /iso3-EGFP without the inclusion of shRNA-mCherry to avoid potential bleed-through of signals. Additionally, we treated the cells with MG132 to assess protein stability.

Our western blot analysis confirmed Reviewer #3's suggestion. In the presence of 5 μ M MG132, the expression levels of TMEM87A-iso1 Δ /iso3-EGFP were increased compared to the absence of MG132. This observation indicates that the Golgi-targeting motif contributes to protein stability by potentially increasing degradation.

Supplementary Fig.1 | f, Western blot analysis depicting the expression levels of EGFP-tagged TMEM87A isoforms (iso1, isoΔ, iso3) in cultured human astrocytes under conditions of both absence and presence of 5μM MG132.

Furthermore, our ICC staining results revealed intriguing findings. With prolonged exposure time, we observed the precise localization of TMEM87A-iso1Δ/iso3-EGFP in the presence of 5μM MG132. Notably, this localization pattern differed from that of TMEM87A-iso1-EGFP, suggesting that the predicted Golgi-targeting motif indeed plays a role in Golgi localization.

In our revised manuscript, we have changed supplementary Fig.1e and 1f and refined the interpretation (p.3, highlighted in blue), as follows:

“We observed distinct and strong fluorescence signal indicating Golgi localization for isoform 1. In contrast, isoform 1Δ and isoform 3 exhibited weak fluorescence signals with different localization, even when the fluorescence intensity was saturated (Supplementary Fig. 1e). In the presence of 5μM MG132, the expression levels of TMEM87A-iso1Δ/iso3-EGFP were increased compared to the absence of MG132 (Supplementary Fig. 1f), indicating that the Golgi-targeting motif contributes to not only Golgi localization but also protein stability by potentially increasing degradation.”

Supplementary Fig.1 | e, Localization of C-terminal EGFP-tagged TMEM87A isoforms (iso1, isoΔ, iso3) in cultured human astrocytes in the absence or presence of 5μM MG132.

3- Figure 1e: please also provide a picture with lower exposure time to detect Golgi-localization of overexpressed TMEM87A in CHO-K1.

We appreciate Reviewer #3's suggestion regarding Figure 1e. In response, we have captured images with varying exposure times, including higher, intermediate, and lower exposure times, of EGFP-tagged TMEM87A transfected CHO-K1 cells.

We understand the reviewer's request for an additional image showing Golgi-localization of overexpressed TMEM87A in CHO-K1 cells with lower exposure time in the manuscript. However, it's important to clarify the intention behind including the fluorescence image of EGFP-tagged TMEM87A overexpressed in CHO-K1 cells in Figure 1e. This image was primarily aimed at showcasing the expression of TMEM87A in the plasma membrane of the heterologously expressed system, particularly for facilitating whole-cell membrane patch experiments.

We share the concern that presenting only the Golgi-localized TMEM87A image might potentially confuse readers by diverting the focus from the intended demonstration of TMEM87A expression in the plasma membrane. Therefore, we believe it would be more appropriate to provide these additional images in the rebuttal section, where we can offer a more comprehensive explanation and context for their inclusion.

In addition, please modify the title of Fig1 as measurement of currents have been performed in CHO-K1 cells (not in astrocytes), if I am not mistaken.

Reviewer #3's comment regarding the title of Figure. 1. Upon review, we acknowledge that the current title is not appropriate for the data presented. In response to Reviewer #3's suggestion, we have revised the title as follows:

"TMEM87A regulates Golgi pH in human astrocytes and mediates voltage- and pH-dependent, inwardly rectifying cationic currents in CHO-K1 cells.

4- Page 30 of the rebuttal letter: the additional information provided by the authors regarding the surface biotinylation answers appropriately to my comments.

We appreciate Reviewer #3's positive assessment of revised manuscript regarding the surface biotinylation.

5- Fig 6d (rebuttal letter pages 33-34):

The co-localization of TMEM87A with the Golgi apparatus is still not convincing. In astrocytes the signal for Golgin97 is very diffuse. In addition, if I am not mistaken, visualization of Golgin-97 has been performed thanks to immunostaining. Consequently, I do not understand while only the cell in the middle of the field shows golgin-97 -positive signal. Golgin-97 is expected to be expressed also in the neighboring cell. The authors might consider fluorescence bleed-through of the GFAP signal to the channel of the Golgi-97 staining. To my opinion, the upper panel of Fig.6 d cannot be published as it is.

We appreciate Reviewer #3's insightful feedback, and we fully understand the concerns raised regarding Fig. 6d.

In response to Reviewer #3's concern, we would like to clarify that Fig. 6d represents a single cell within the field of view. To address this concern, we have replaced it with an image that has a wider scale, allowing observation of surrounding cells. In the revised figure, Golgin-97 expression is observed in neighboring astrocytes, it may not be apparent in all cells due to the heterogeneity of cell types present in the hippocampal *stratum radiatum*. This heterogeneity includes both astrocytes (GFAP positive, indicated by gray arrows) and inhibitory neurons (GFAP negative, indicated by yellow arrowheads). Consequently, Golgin-97 expression is expected to be expressed in astrocytes within the hippocampal *stratum radiatum*, consistent with our observations.

Fig.6 | d, Colocalization of GolpHCat with Golgin-97 or Giantin in hippocampal astrocyte (GFAP) and neuron (NeuN) of WT mice, respectively. Gray arrows indicate GFAP-positive cells, and yellow arrowheads indicate GFAP-negative cells.

Regarding the concern about fluorescence bleed-through, we have thoroughly examined our imaging protocols and confirmed that there is no significant bleed-through effect observed in our experiments. Although Golgin-97 signal may appear similar to GFAP signal in some astrocytes, Golgin-97 expression is in general distinct from GFAP signal and not a result of fluorescence bleed-through.

6- Minor point: page 6 of the revised manuscript: Is the reference to Extended Data Fig.5f the right one ?

We sincerely apologize for this error and appreciate Reviewer #3's careful peer-review. In response to this valuable feedback, we have fixed the errors as follows:

“On page 6, “Extended Data Fig. 5f” has been corrected to “Supplementary Fig. 4f”

The authors appropriately modified the revised version to take into account the other concerns that I raised after reviewing the initial version of the manuscript.

We sincerely appreciate Reviewer #3's overall positive evaluation of our work.

Reviewer #1 (Remarks to the Author):

In the revised version of the manuscript, the authors have appropriately addressed and implemented all comments and suggestions. The manuscript is a great example of a project in which a scientific question has been addressed using many different approaches and techniques.

Reviewer #3 (Remarks to the Author):

The authors answered to my comments.
I recommend publication in Nature Communications.

REVIEWERS' COMMENTS

Reviewer #1 (Remarks to the Author):

In the revised version of the manuscript, the authors have appropriately addressed and implemented all comments and suggestions. The manuscript is a great example of a project in which a scientific question has been addressed using many different approaches and techniques.

Thank you sincerely for your thoughtful and valuable feedback on our research manuscript. Your suggestions and insights were immensely helpful in improving the paper.

Reviewer #3 (Remarks to the Author):

The authors answered to my comments.
I recommend publication in Nature Communications.

We are deeply grateful for your insightful comments and recommendations. Thanks to your positive evaluation, our paper has been accepted for publication in Nature Communications. Thank you.